# Application of *Ligilactobacillus salivarius* CECT5713 to Achieve Term Pregnancies in Women with Repetitive Abortion or Infertility of Unknown Origin by Microbiological and Immunological Modulation of the Vaginal Ecosystem

**DOI:** 10.3390/nu13010162

**Published:** 2021-01-06

**Authors:** Leónides Fernández, Irma Castro, Rebeca Arroyo, Claudio Alba, David Beltrán, Juan M. Rodríguez

**Affiliations:** 1Department of Galenic Pharmacy and Food Technology, Complutense University of Madrid, 28040 Madrid, Spain; 2Department of Nutrition and Food Science, Complutense University of Madrid, 28040 Madrid, Spain; irmacastro@ucm.es (I.C.); rebecaa@vet.ucm.es (R.A.); c.alba@ucm.es (C.A.); 3Centro de Diagnóstico Médico, Ayuntamiento de Madrid, 28006 Madrid, Spain; beltrangine@gmail.com

**Keywords:** infertility, repetitive abortion, implantation failure, *Lactobacillus salivarius*, probiotics, vaginal microbiome, TGF-β, VEGF

## Abstract

In this study, the cervicovaginal environment of women with reproductive failure (repetitive abortion, infertility of unknown origin) was assessed and compared to that of healthy fertile women. Subsequently, the ability of *Ligilactobacillus salivarius* CECT5713 to increase pregnancy rates in women with reproductive failure was evaluated. Vaginal pH and Nugent score were higher in women with reproductive failure than in fertile women. The opposite was observed regarding the immune factors TGF-β 1, TFG-β 2, and VEFG. Lactobacilli were detected at a higher frequency and concentration in fertile women than in women with repetitive abortion or infertility. The metataxonomic study revealed that vaginal samples from fertile women were characterized by the high abundance of *Lactobacillus* sequences, while DNA from this genus was practically absent in one third of samples from women with reproductive failure. Daily oral administration of *L. salivarius* CECT5713 (~9 log_10_ CFU/day) to women with reproductive failure for a maximum of 6 months resulted in an overall successful pregnancy rate of 56%. The probiotic intervention modified key microbiological, biochemical, and immunological parameters in women who got pregnant. In conclusion, *L. salivarius* CECT5713 has proved to be a good candidate to improve reproductive success in women with reproductive failure.

## 1. Introduction

Increasing evidence has highlighted the relevance of the microbiota of the female genital tract for human reproduction [1,2]. Under physiological conditions, and in contrast to the gut, the human vaginal microbiota is usually characterized by a low microbial diversity and the dominance of bacteria from the genus *Lactobacillus* [3,4]. In fact, a low diversity in the gut has been linked to a variety of gastrointestinal processes, including inflammatory bowel disease [5], while a high diversity in the vagina has been associated to vaginosis [6].

The vaginal microbiota in healthy reproductive-age women is mainly composed of one or a few *Lactobacillus* species, which represent more than 90% of the total microbiota [7,8]. In a seminal study, the bacterial communities of 396 asymptomatic women were classified into five distinct vaginotypes; four of them were dominated by *Lactobacillus crispatus*, *Lactobacillus gasseri*, *Lactobacillus iners*, and *Lactobacillus jensenii*, respectively; in contrast, the fifth one had lower proportions of lactobacilli and was predominantly composed of strictly anaerobic bacterial genera, such as *Gardnerella*, *Prevotella*, *Megasphaera*, *Atopobium,* or *Dialister* [3]. This last vaginotype was associated to high Nugent scores, a Gram-staining based technique used for the diagnosis of bacterial vaginosis (BV).

Several factors are known to contribute to interindividual and intraindividual changes in the vaginal microbiota [9]. Although shifts between different vaginotypes may occur naturally, increase of diversity and colonization by strict anaerobes and decrease or depletion of lactobacilli are considered as risk factors for BV. In fact, vaginal microbiota dysbiosis has been associated with higher rates of intra-amniotic infection, premature delivery, spontaneous abortion, and infertility [10,11,12,13,14,15].

Different studies have shown that infertile women harbor a differential vaginal microbiota when compared to fertile women [16,17,18,19]. Therefore, the composition of the vaginal microbiota (and, particularly, any deviation from the *Lactobacillus*-dominated, low-diversity vaginal microbiome) may play a key role in fertility and in the outcomes of assisted reproduction treatments (ARTs) [20,21,22]. Abundant isolation of enterococci, streptococci, staphylococci, and/or Gram-negative bacteria (*Escherichia coli*, *Klebsiella pneumoniae*) from the tip of the catheter used for embryo transfer has been correlated with lower implantation and pregnancy rates and increased miscarriage rates [23], while abundant isolation of lactobacilli and low density or no isolation of the aforementioned bacteria has been correlated with better reproductive outcomes [24,25,26,27,28]. Metataxonomic studies of endometrial samples have also revealed that an abnormal endometrial bacterial profile (with a low percentage of sequences of the genus *Lactobacillus*) is a common feature in a high percentage of infertile women subjected to ART [21,29]. Although at least a part of the bacterial DNA detected in endometrial samples may arise from vaginal contamination during sampling, these studies suggest that an abundant presence of *Lactobacillus* DNA in such samples may be a predictor of implantation success [29,30].

As a consequence, the assessment of the microbial communities in the reproductive tract should be considered as a relevant part of the evaluation and personalized care in cases of reproductive failure of unknown cause or origin. When this happens, the use of probiotics may be a possible strategy to modulate the reproductive tract microbiome and to increase the success rates [31]. However, such a combined strategy (assessment of vaginal communities together with use of a target-selected probiotic) has not been explored yet, and commercially available probiotics are being empirically prescribed for repopulation of the female reproductive tract with *Lactobacillus* strains [2], without a proper scientific evidence of their actual usefulness.

Lactobacilli may have different biological activities that contribute to fertility and to a healthy pregnancy, including, among others: (a) the inhibition of the colonization and growth of potentially harmful microbes, including viruses, bacteria, yeast, and protozoa that may compromise fertility [32,33]; (b) contribution to angiogenesis and vasculogenesis that may favor the implantation of the embryo [34]; and, (c) induction of immunomodulation activities, such as those involved in implantation and in tolerance towards the embryo, first, and the fetus, later [35,36]. However, those properties might be strain-specific and, therefore, a strain-by-strain evaluation has to be performed for this specific target.

*Lactobacillus salivarius* CECT5713 [37] has been shown to be a probiotic strain suitable for applications in the mother–infant dyad due to a wide repertoire of desirable phenotypic and genotypic properties [38]. This includes a high survival rate when exposed to gastrointestinal tract conditions, a high acidifying activity, and antimicrobial, anti-inflammatory, and immunomodulatory properties, which have been demonstrated *in vitro*, in animal models, and in human clinical trials [38,39,40,41,42,43,44]. Therefore, and after evaluating some vaginal-related properties in this study, it was selected to be administered in a clinical trial in order to assess its efficacy for the infertility target. It must be highlighted that this species has been renamed as *Ligilactobacillus salivarius* in the recent proposal for reclassification of the genus *Lactobacillus* [45].

In this context, the first objective of this study was to assess the differences in several vaginal parameters (pH, Nugent score, microbiota composition as determined through culture and metataxonomic methods, and soluble immune factor levels) between women with reproductive failure (because of repetitive abortion during the first 12 weeks of pregnancy or infertility of unknown origin) and fertile women. The second objective was to evaluate the ability of *L. salivarius* CECT5713 to modulate those vaginal parameters and to increase pregnancy rates (currently ~29% after IVF procedures in this setting) in the group of women with reproductive failure.

## 2. Materials and Methods

### 2.1. Characterization of Vaginal-Related Properties of L. salivarius CECT5713

An overlay method [46] was used to determine the ability of *L. salivarius* CECT5713 to inhibit the growth of various species of bacteria and yeasts. It was performed as described previously [37]. All indicator strains had been previously isolated from clinical cases of vaginal or cervical infections, and included five strains of *G. vaginalis*, three of *Streptococcus agalactiae* and of *Candida albicans*, and two of *Candida glabrata*, *Candida parapsilosis,* and *Ureaplasma urealyticum* (our own culture collection). All inhibitory activity assays were performed in triplicate.

The ability of *L. salivarius* CECT5713 to aggregate with cells of the indicator strains cited above was investigated following the procedure of Younes et al. [47]. The suspensions were observed under a phase-contrast microscope. Adherence to vaginal epithelial cells collected from healthy premenopausal women was performed and interpreted as described previously [48]. Adherence was measured as the number of lactobacilli adhered to the vaginal cells in 20 random microscopic fields. *L. salivarius* CECT9145 was used as a control strain because of its high adherence to vaginal cells [49]. The assay was performed in triplicate.

Initially, the α-amylase activity of *L. salivarius* CECT5713 was qualitatively assessed using the method described by Padmavathi et al. [50]. Briefly, the strain was inoculated into a modified MRS media containing starch (0.5% peptone, 0.7% yeast extract, 0.2% NaCl, 2% starch, and 1.5% agar). The plates were incubated at 37 °C for 48 h in anaerobiosis and, then, the zone of clearance was observed by adding Gram’s iodine as detecting agent. Quantitation of the cell-bound α-amylase activity of *L. salivarius* CECT5713 was done with a kit (Kikkoman Co., Tokyo, Japan) using 2-chloro-4-nitrophenyl 6^5^-azido-6^5^-deoxy-β-maltopentaoside as substrate and using conditions described previously [51]. One unit of activity was defined as the amount of enzyme needed to release 1 μmol 2-chloro-4-nitrophenol from 2-chloro-4-nitrophenyl 6^5^-azido-6^5^-deoxy-β-maltopentaoside per min at 37 °C.

### 2.2. Participants, Sampling, and Design of the Human Study

A total of 58 women, aged 28–45, participated in this study (Table 1). Volunteers were classified into 3 groups. All women in the RA group (*n* = 21) had a history of recurrent miscarriage with three or more pregnancy losses during the first 12 weeks of pregnancy. All women of the INF group (*n* = 23) had a history of infertility (inability to conceive) despite being the recipients of ART for at least three times, including two cycles, at least, of in vitro fertilization (IVF). Finally, the control group (*n* = 14) included fertile women having at least two children after uncomplicated term pregnancies. None of the women of the RA and INF groups received ART during the whole period of the study. None of the RA group components were diagnosed of antiphospholipidic syndrome and, therefore, they did not receive either heparin and/or salicylic acid during the study. None of the participants had received hormonal therapy, antibiotics or probiotics in the 4 weeks previous to sampling. Vaginal samples were taken at least 7 days after coitus to avoid or minimize the impact of the partner’s semen on the vaginal pH, microbiota composition or immunoprofile (in the latter case, particularly in relation to the concentration of the two isoforms of the transforming growth factors beta 1 and 2 (TGF-β 1 and TGF-β 2)). Women with lactose intolerance or cow’s milk protein allergy were excluded because of the excipient used to administer the strain in the subsequent pilot trial (see below). Informed consent was obtained from all subjects involved in the study.

At recruitment (within the first three days post-ovulation; day 0), two samples were collected: A vaginal swab specimen for in fresh determination of the Nugent score, and a cervicovaginal lavage (CVL) of the cervical and the vaginal walls with 10 mL of sterile normal saline for all the other analysis. Aliquots of the CVL samples were used for culture-based analysis. Subsequently, CVL samples were clarified by centrifugation at 800× *g* for 10 min at 4 °C. Aliquots of CVL supernatants and cell pellets were stored at −80 °C until the immunological and metataxonomic analyses were performed. Demographic, anthropometric, and health data (including a past or present history of recurrent infections at different body locations and use of antibiotics) were recorded at recruitment (Appendix A). High use of antibiotics was defined as receiving ≥4 antibiotic treatments per year because of recurrent infections while a range between 0 and 2 annual treatments was considered as a low use of antibiotics.

Starting at day 0, women of the RA and INF groups consumed (oral route) a daily sachet with ~50 mg of freeze-dried probiotic (~9 log_10_ CFU of *L. salivarius* CECT5713) for 6 months or until a diagnosis of pregnancy (whatever happened first). At that point, the same two samples described above were collected from each woman. After a diagnosis of pregnancy, oral administration of the probiotic strain was maintained until the 15th week of pregnancy. All the spontaneous pregnancies that occurred within the first year after day 0 were recorded in this study.

Probiotic-containing sachets were kept at 4–8 °C throughout the study. All volunteers signed a written consent and were provided with diaries to record compliance with the study product intake. Minimum compliance rate (% of the total treatment doses) was set at 86%. This study was conducted according to the guidelines laid down in the Declaration of Helsinki and it was approved by the Ethical Committee of Biomedical Research of Consejería de Salud y Familias (Junta de Andalucía, Granada, Spain) (P050/19, Act 11/19). The study was registered in the ClinicalTrials.gov database (NCT04446572).

### 2.3. Measurement of Vaginal pH and Nugent Score

At each of the two study visits, the pH of the lateral vaginal wall was measured (Whatman pH paper, pH 3.8–5.5 and pH 6.0–8.1). Nugent scoring was performed as described previously [52]. Briefly, the swab material was transferred to a glass slide, heat fixed, and Gram stained. Gram-positive, Gram-negative, and Gram-variable bacterial morphotypes were quantified. A Nugent score of 0–3 was considered normal, 4–6 was considered intermediate, and 7–10 was considered consistent with bacterial vaginosis [52].

### 2.4. Culture-Dependent Analysis

CVL samples collected during the trial were serially diluted and plated onto Columbia Nalidixic Acid (CNA), Gardnerella (GAR), CHROMagar StrepB (CHR), Mac Conkey (MCK), Mycoplasma (MYC), and Sabouraud Dextrose Chloramphenicol (SDC) agar plates (BioMerieux, Marcy l’Etoile, France) for selective isolation and quantification of the main cultivable non-*Lactobacillus* bacteria and yeasts that may be found in the vagina, including the agents most frequently involved in vaginal infections. They were also inoculated onto agar plates of MRS (Oxoid, Basingstoke, UK) supplemented with either L-cysteine (2.5 g/L) (MRS-C) or horse blood (5%) (MRS-B) for isolation of lactobacilli, including *L. iners* (MRS-B). All media were incubated for 48 h at 37 °C under aerobic conditions, with the exception of the MRS-C and MRS-B plates, which were incubated anaerobically (85% nitrogen, 10% hydrogen, 5% carbon dioxide) in an anaerobic workstation (DW Scientific, Shipley, UK) for up to 72 h. After incubation, the colonies were recorded and at least one representative of each colony morphology was selected from the agar plates. The isolates were identified by Matrix Assisted Laser Desorption Ionization-Time of Flight (MALDI-TOF) mass spectrometry (Bruker, Bremen, Germany). When the identification by MALDI-TOF was not possible at the species level (particularly in the case of lactobacilli isolates), the identification was carried out by 16S ribosomal RNA (rRNA) gene sequencing as described by Mediano et al. [53].

### 2.5. DNA Extraction from the Samples

Approximately 1 mL of each CVL sample was used for DNA extraction following a method described previously [54]. Extracted DNA was eluted in 22 μL of nuclease-free water and stored at −20 °C until further analysis. Purity and concentration of each extracted DNA was initially estimated using a NanoDrop 1000 spectrophotometer (NanoDrop Technologies, Inc., Rockland, DE, USA). Negative controls (blanks) were processed in parallel.

### 2.6. Real-Time Quantitative PCR (qPCR) Assay for the Detection and Quantification of L. salivarius DNA

Primers and conditions for quantification of *L. salivarius* DNA have been described previously [55]. The DNA concentration of all samples was adjusted to 5 ng µL^−1^. A commercial real-time PCR thermocycler (CFX96™, Biorad Laboratories, Hercules, CA, USA) was used for all experiments. Standard curves using 1∶10 DNA dilutions (ranging from 2 ng to 0.2 pg) from *L. salivarius* CECT5713 were used to calculate the concentrations of the unknown bacterial genomic targets. Threshold cycle (Ct) values between 15.29 and 20.07 were obtained for this range of *L. salivarius* DNA (R^2^ = 0.9915). The Ct values measured for DNA extracted from non-target species (*L. reuteri* MP07 and *Lactobacillus plantarum* MP02; our own collection) were ≥39.27 ± 0.64. These two control strains were selected because they belong to the *L. salivarius* taxonomically closest species [56]. All samples and standards were run in triplicate.

### 2.7. Metataxonomic Analysis

The V3-V4 hypervariable region of the 16S rDNA was amplified by PCR using the universal primers S-D-Bact-0341-b-S-17 (CCTACGGGNGGCWGCAG) and S-D-Bact-129 0785-a-A-21 (GACTACHVGGGTATCTAATCC) [57] and sequenced in the MiSeq system of Illumina at the facilities of Parque Científico de Madrid (Tres Cantos, Spain). Barcodes appended to 3′ and 5′ terminal ends of the PCR amplicons allowed separation of forward and reverse sequences in a second PCR-reaction. DNA concentration of the PCR products was quantified in a 2100 Bioanalyzer system (Agilent, Santa Clara, CA, USA). After pooling the PCR products at about equal molar ratios, DNA amplicons were purified by using a QIAEX II Gel Extraction Kit (Qiagen, Hilden, Germany) from the excised band having the correct size after running on an agarose gel. DNA concentration was then quantified with PicoGreen (BMG Labtech, Jena, Germany). The pooled, purified and barcoded DNA amplicons were sequenced using the Illumina MiSeq pair-end protocol (Illumina Inc., San Diego, CA, USA) following the manufacturer’s protocols.

### 2.8. Bioinformatic Analysis

Raw sequence data were demultiplexed and quality filtered using Illumina MiSeq Reporter analysis software. Microbiome bioinformatics was done with QIIME 2 2019.1 [58]. Denoising was performed with DADA2 [59]. Taxonomy was assigned to ASVs using the q2-feature-classifier [60] and the naïve Bayes classifier *classify-sklearn* against the SILVA database version 132 [61]. Posterior bioinformatic analysis was conducted using the R version 3.5.1 (https://www.R-project.org) [62]. A table of Operational Taxonomic Units (OTUs) counts per sample was generated, and bacterial taxa abundances were normalized to the total number of sequences in each sample. The relative abundance values of the different bacterial taxa in the three groups of CVL samples (control, RA and INF) were analyzed using the linear discriminant analysis (LDA) effect size (LEfSe) algorithm [63] in an online version (http://huttenhower.sph.harvard.edu/galaxy/). Alpha diversity was studied with the Shannon and Simpson diversity indexes with the R Vegan package (Version 2.5.6) (https://github.com/vegandevs/vegan/). Beta diversity was studied using principal coordinates analysis (PCoA) to visually display patterns of bacterial profiles at the genus level through a distance matrix containing a dissimilarity value for each pairwise sample comparison. The Bray–Curtis and binary Jaccard indices were used for quantitative (relative abundance) and qualitative analyses (presence/absence), respectively. Analysis of variance of the distance matrices was performed with the “nonparametric *MANOVA* test” Adonis with 999 or permutational multivariate ANOVA (PERMANOVA) with 999 permutations with the R Vegan package. The heatmap graph was generated by using *gplots* package. Dendogram linkages were based on the relative abundance of the 20 most abundant bacterial genera within the samples and on the complete linkage method for hierarchical clustering (*hclust* function).

### 2.9. Immunological Analysis

The concentrations of several soluble immune factors (IL1β, IL1ra, IL2, IL4, IL5, IL6, IL7, IL8, IL9, IL10, IL12, IL13, IL15, IL17, IL6, basic FGF, eotaxin, GCSF, GMCSF, IFNγ, MCP1, MIP1α, MIP1β, PDGF-BB, RANTES, TNFα, VEGF) were determined by magnetic bead-based multiplex immunoassays, using a Bioplex 200 instrument (Bio-Rad, Hercules, CA, USA) and the Bio-Plex Pro™ Human Cytokine 27-plex Assay (ref. M500KCAF0Y, Bio-Rad). In parallel, the levels of TGF-β 1 and TGF-β 2 were measured by ELISA with the RayBio^®^ Human TGF-β 1 and Human TGF-β 2 ELISA kits, respectively (RayBiotech, Norcross, GA, USA). All determinations were carried out following the manufacturer’s protocols and standard curves were performed for each analyte.

### 2.10. Statistical Analysis

Microbiological data were recorded as CFU/mL and transformed to logarithmic values before statistical analysis. The normality of data distribution was analyzed using the Shapiro–Wilks test. Then, the quantitative variables were expressed as means and 95% confidence intervals (CI) or standard deviations (SD) when normally distributed and as medians and interquartile ranges (IQR) if they did not follow a normal distribution. The qualitative values were presented as total number of events and percentages. One-way ANOVA tests were used to compare the means of the experimental groups and Scheffé post hoc tests were used to identify which pairs of means were statistically different. The effect of the probiotic intervention on several vaginal parameters in each group of women with reproductive failure was analyzed using one-way ANOVA repeated measures tests. The Fisher’s exact probability test, or the Freeman–Halton extension of the Fisher exact probability test for a 2 × 3 contingency table, was used for comparison of proportions and frequencies. For non-parametric analyses, differences between groups were assessed using Kruskal–Wallis tests and Wilcoxon–Mann–Whitney tests to identify which pair of groups were different, with Bonferroni correction for multiple comparisons when indicated. Correlations between the 20-major relative abundant bacterial genera were visualized using R package *qgraph* [64]. Statistical analysis and plotting were performed either using Statgraphics Centurion XVIII version 18.1.06 (Statgraphics Technologies, Inc., The Plains, VA, USA) or in the R environment (version 3.5.1; R-project, http://www.r-project.org) and *ggplot2* [Wickham, 2016]. Differences were considered statistically significant at *p* < 0.05. 

## 3. Results

### 3.1. Characterization of Vaginal-Relevant Properties of L. salivarius CECT5713

*L. salivarius* CECT5713 showed inhibitory antimicrobial activity (inhibition zone > 2 mm around the streak) against all the *G. vaginalis*, *S. agalactiae, C. albicans*, *C. glabrata*, *C. parapsilosis,* and *U. urealyticum* strains used as indicators in this study. The strain was able to form large, well defined co-aggregates with all the selected vaginal and cervical pathogens. Co-aggregation was particularly intense with *G. vaginalis*, *S. agalactiae,* and *C. albicans* strains. In this study, the strain tested was strongly adhesive to vaginal epithelial cells, a mean (±SD) of 329 (±46) adherent lactobacilli in 20 random microscopic fields. The mean (±SD) value for *L. salivarius* CECT9145, a control strain with a high adherence to vaginal cells, was 336 (±52) adherent lactobacilli in 20 microscopic fields. Extracellular amylase production by *L. salivarius* CECT5713 was observed by the zone of clearance around the colonies (~2.0 mm) when flooded with iodine solution. Later, when the α-amylase activity was measured, this strain showed a high level of α-amylase activity (0.83 U/mL) at 16 h (concentration of *L. salivarius* CECT5713: ~8.6 log_10_ CFU/mL), and could be detected in supernatants at a similar level for up to 48 h (when the assay was finished).

### 3.2. Demographic, Anthropometric, and Clinical Characteristics of the Participants in the Human Study

The characteristics of the 58 women that participated in this study are presented in Table 1. The mean (95% CI) age in the control group was 34.6 years (33.5–35.8), while in those of repetitive abortions (RA) and with infertility of unknown origin (INF) was 39.4 (38.5–40.4) and 38.0 (37.1–38.9) years, respectively (Table 1). Women in the control group were significantly younger than other participants (*p* < 0.001; one-way ANOVA), but there were no differences in mean values of body weight and height between the three groups of women (Table 1).

About 71% of the women in the control group had a regular menstrual cycle, while in the other two (RA and INF) this percentage was 48%, although this difference was not statistically significant (*p* = 0.337; Fisher exact probability tests). No differences were observed in the mean duration of the menstrual cycle that was 28, 27.4, and 27.5 days for women in the control, RA, and INF groups, respectively (Table 1).

Interestingly, statistically significant differences were found between the control women and those in the other two groups regarding a history of recurrent vaginal and urinary tract infections (*p* = 0.017 and *p* = 0.006, respectively; Fisher exact probability tests) and the use of antibiotics both during infancy (*p* < 0.001) and adulthood (*p* = 0.003), which were higher in the last two groups (Table 1; Appendix A). A trend to a higher rate of ORL infections (pharyngitis, otitis) among women with repetitive abortion or infertility was also observed but it did not reach statistical significance (*p* = 0.057). In contrast, no differences were observed among the three groups in relation to the rates of skin, lower respiratory tract and gastrointestinal infections (Table 1).

### 3.3. Baseline Vaginal Health Parameters

The vaginal pH values of the control group (4.53; range 4.38–4.68) were statistically different from those of the two study groups: 5.67 (5.55–5.79) and 5.96 (5.84–6.07) for RA and INF, respectively (*p* = 0.000; one-vay ANOVA). Similarly, the Nugent scores of the two study groups were significantly higher (5.95 (5.54–6.37) and 6.30 (5.91–6.70), respectively), than those from controls (1.79 (1.27–2.30); *p* = 0.000; one-way ANOVA) (Table 2). The CVL concentrations of the growth factors TGF-β 1, TFG-β 2 and VEFG of the control group were 4.83 (4.65–5.01) pg/mL, 3.22 (3.10–3.34) pg/mL, and 406.0 (322.0–490.0) pg/mL, respectively, while they appeared to be halved in both study groups (RA and INF), the differences being statistically significant (Table 2). No differences were observed among the three groups in relation to the remaining soluble immune factors analyzed in this work, which showed a high degree of interindividual variability (data not shown).

All women of the control group harbored lactobacilli in their vaginas (*n* = 14), the mean (95% CI) value being 7.24 (6.89–7.60) log_10_ CFU/mL using culture-dependent assessment. The frequency of lactobacilli detection was lower in the RA and INF groups: 57% and 26%, respectively (*p* < 0.001; Fisher exact probability tests). In addition, mean lactobacilli concentrations were 2.20 and 1.46 log_10_ units lower in CVL samples from lactobacilli-positive women in the RA and INF groups, respectively. The lactobacilli profile was also different (Figure 1). Seven species were identified in the samples from women of the control group, including *L. crispatus* (the dominant species), *L. jensenii*, *L. gasseri*, *L. iners*, *Limosilactobacillus* (formely *Lactobacillus*) *fermentum*, *L. salivarius*, and *Limosilactobacillus vaginalis.* However, the lactobacilli species profiles in the study groups (RA and INF) were narrower than in controls and *L. fermentum*, *L. salivarius*, and *L. vaginalis* were not detected. *L. crispatus* was the dominant species in 6 samples (43%) from fertile women, 5 samples (24%) from women with repetitive abortion and only 1 sample (4%) from infertile women. It is interesting to note that *L. iners* was isolated only from one CVL sample of the control group while it was isolated from about one-third (5 out of a total of 18 lactobacilli positive samples) from samples of RA and INF groups. *L. salivarius* was detected in the sample of a unique woman from the control group as determined by species-specific qPCR (7.29 log_10_ copies/mL) and culture (7.3 log_10_ CFU/mL) (Table 2). The strain was genetically different from *L. salivarius* CECT5713 (results not shown).

Globally, the comparison of RA and INF groups at the beginning of the study revealed some statistically relevant differences (Figure 2). The mean of the vaginal pH values was 0.29 units higher in the INF group, but the opposite was observed for TGF-β 1 and VEGF, which had mean concentrations 0.43 pg/mL and 94 pg/mL higher, respectively, in the RA group. No differences were observed regarding other characteristics, including age, weight, height, Nugent score, TGF-β 2, and lactobacilli viable counts (Figure 2).

The 16S rRNA gene sequencing analysis of the CVL samples (*n* = 58) yielded 4,363,364 high quality filtered sequences, ranging from 33,160 to 139,044 per sample (median [IQR] = 73,383 [66,587–82,821] sequences per sample). Sequences were assigned to a total of 23 phyla and 453 genera, and Figure 3 shows the 5 most abundant phyla and the 20 most abundant genera in CVL samples from the fertile control group and from the RA and INF groups. The comparison of the relative abundance (% of total) of sequences at the phylum level from the three groups revealed statistically significant differences with regard to the 4 dominant phyla: *Firmicutes*, *Actinobacteria*, *Proteobacteria,* and *Bacteroidetes* (Table 3). The most frequent (present in all samples) and abundant phylum was *Firmicutes* (Figure 3). The relative abundance of *Firmicutes* in samples provided by fertile controls (median [IQR] = 99.60% [99.18–99.80%]) was higher than in samples from women of RA and INF groups (median [IQR] = 97.29% [72.34–99.35%] and 89.96% [52.46–98.85%], respectively) (*p* < 0.001; Kruskal–Wallis rank test with Bonferroni correction) (Table 3). In contrast, the median (IQR) values of the relative abundance of *Actinobacteria, Proteobacteria,* and *Bacteroidetes* were higher in women of the RA and INF groups (*p* < 0.012, *p* < 0.003, and *p* < 0.006, respectively; Kruskal–Wallis rank tests with Bonferroni correction) (Table 3).

The only bacterial genus that was detected in all samples was *Lactobacillus*, but there were significant differences in its relative abundance in samples from the three groups (Table 3; Figure 3). The median [IQR] relative abundance of *Lactobacillus* in CVL samples from women of RA and INF groups (93.49% [67.18–97.53%] and 71.95% [0.76–94.09%], respectively) was lower than in samples from fertile control women (97.88% [96.92–99.31%]) (*p* = 0.001; Kruskal–Wallis rank test with Bonferroni correction) (Table 3). In fact, the only bacterial genus that characterized and differentially explained the greatest difference between the microbial communities in CVL samples between fertile control women and women of RA and INF groups was *Lactobacillus*, according to the LEfSe analysis (Figure 4).

Other genera were present in a variable number of samples, ranging from 96% (*Staphylococcus* in the INF group) to 7% (*Escherichia*/*Shigella* in the control group), but the median relative abundance of any of these genera was <1% (Table 3). The bacterial profile at the genus level in some individual samples from women in the RA and INF groups did not differ from that of samples from women from the fertile control group, which were highly homogenous (Figure 5). However, aberrant profiles with reduced content or even complete absence of *Lactobacillus* were registered in some samples from women of the RA and INF groups (Figure 5).

The analysis of alpha diversity at the genus level, calculated either by the Shannon or the Simpson’s indices, revealed significant differences between the vaginal microbiota of women in the fertile and INF groups (*p* < 0.001; Kruskal–Wallis tests with Bonferroni correction) (Figure 6A,B).

The analysis of the beta diversity, calculated according to the relative abundance of bacterial genera (Bray–Curtis distance) and the presence/absence of bacterial genera (Binnary Jaccard distance matrix), indicated that the profiles of bacterial genera of CVL samples of the 3 groups clustered apart (*p* = 0.004 and *p* = 0.002, respectively; PERMANOVA) (Figure 6C,D). In addition, samples from fertile controls clustered closer (shorter distance to centroid) according to the relative abundance of bacterial genera (Bray–Curtis distance) than those from RA and INF groups, indicating that the bacterial profiles in CVL samples from controls were highly uniform (Figure 6E,F).

An initial assessment of potentially dominant patterns in the bacteriological profile of the CVL samples is shown in the heatmap plot presented in Figure 6G. There was a clear separation of samples based on the presence of *Lactobacillus*. One cluster was characterized by the marked and almost exclusively presence of *Lactobacillus* in CVL samples. This cluster comprised all the samples from fertile women although not exclusively, because it included also some samples from the RA and INF groups. The second cluster was characterized by the absence or reduced presence of *Lactobacillus* and the presence of multiple bacterial genera, such as *Gardenella* and *Bifidobacterium*. This second cluster contained exclusively CVL samples from the RA and INF groups. Although globally there was no clear separation between the CVL samples from the three groups, it was perceived a higher similarity between samples from the fertile control group and women with a history of repetitive abortion than between the fertile control group and women with infertility of unknown origin (Figure 6G).

### 3.4. Main Outcome of the Clinical Trial: Pregnancies and Successful Pregnancies

Administration of *L. salivarius* CECT5713 (~9 log_10_ CFU/day) for 6 months (or until a diagnosis of pregnancy if this happened first) to the women of the RA and INF groups led to 29 pregnancies out of the 44 participating patients. This means a pregnancy effectiveness of 66% with a 95% CI of 52–80% (Table 4). Among them, there were 25 successful pregnancies and 4 abortions. This means an effectiveness for reproductive success of 57% with a 95% CI of 42–72% (Table 4). Interestingly, all successful pregnancies led to full-term singletons (gestational age ≥ 38 weeks).

Women of the RA group had the highest rate of reproductive success (15 full term pregnancies and 2 abortions out of 21 participants) (Table 4). The rate in the INF group was lower although still noticeable: 12 pregnancies (10 full term and 2 abortions) out of 23 enrolled. Therefore, the pregnancy effectiveness and successful pregnancy rates (95% CI) tended to be higher in RA group that in INF group (RR [95% CI] = 1.55 [1.00–2.42] and 1.64 [0.96–2.82], respectively), although the difference between both groups did not reach statistical significance (Table 4). It must be highlighted that all women of these groups had been unsuccessfully subjected to ART interventions in previous attempts to avoid spontaneous miscarriage (RA group) or to get pregnant (INF group).

### 3.5. Secondary Outcomes Associated with the Probiotic Treatment: RA Group

There were no differences in age, weight, or height between women in the RA group that ended up having a successful pregnancy (*n* = 15) and those who did not (*n* = 6) after the probiotic intervention. However, differential changes in their vaginal parameters were observed (Table 5). The vaginal pH of women who delivered was about 0.9 units lower than in those who did not (*p* < 0.001; one-way ANOVA). Similar results were noted for the Nugent score (a mean [95% CI]) reduction of 3.33 [3.73–2.93] units in women who got pregnant after the probiotic intervention versus a mean [95% CI] reduction of 0.67 [1.29–0.04] units in those who did not complete a full-term pregnancy; *p* = 0.000 one-way ANOVA) (Table 5, Appendix A). In fact, the probiotic treatment did not modify the Nugent score in those women that did not get pregnant (*p* = 0.102; one-way repeated measures ANOVA) (Table 5).

The vaginal cytokine concentrations also differed in both subgroups of women (with successful pregnancy or not) in the RA group after the probiotic treatment. There was no modification in the vaginal TGF-β 1, TGF-β 2, and VEGF concentrations with respect to the baseline in the women who did not become pregnant, but there was a mean (95% CI) significant increase of 1.40 (1.18–1.62) pg/mL, 1.25 (1.12–1.38) pg/mL, and 402 (319–485) pg/mL, respectively, in those who did (*p* = 0.000; one-way repeated measures ANOVA) (Table 5, Appendix A). In addition, it should be noted that there were already differences in the concentration of these cytokines even before starting the treatment between those that became and those that did not become pregnant (Table 5).

On the other hand, the probiotic treatment resulted in a mean (95% CI) increase in lactobacilli counts of 2.12 (1.66–2.59) log_10_ CFU/mL in women that finally got pregnant, but there was no change in those that did not (Table 5, Appendix A). The presence of *L. salivarius* (mean [95% CI] = 6.85 [6.58–7.12] log_10_ copies/mL) was confirmed by qPCR in all women that got pregnant, but only in 50% of the women with unsuccessful pregnancies, their concentration being significantly lower (mean [95% CI] = 2.63 [0.41–3.24] copies/mL) (Table 5). The lactobacilli profile in CVL samples obtained at the beginning of the probiotic treatment and after 6 months or until a diagnosis of pregnancy is presented in Figure 7. The most noticeable difference was the presence of viable *L. salivarius* in most women (17/21) after the probiotic treatment. In addition, *L. iners*, which was present in 3 women at the beginning of the study, was isolated at the end of the treatment only from 2 women who did not end up in pregnancy. There were no differences in the metataxonomic profile at the genus level of CVL samples from women of the RA group regarding the pregnancy outcome (Figure 5; Appendix A).

### 3.6. Secondary Outcomes Associated with the Probiotic Treatment: INF Group

The women in the INF group that got pregnant after the probiotic intervention (*n* = 10) and those who did not (*n* = 13) did not differ in age, weight and height. The CVL pH and the Nugent score decreased significantly in all members of the INF group after the probiotic treatment (*p* < 0.05; one-way repeated measures ANOVA), although the magnitude of the change was smaller in the women that did not get pregnant when compared to those that got pregnant (Table 6; Appendix A). Specifically, the mean (95% CI) reductions in CVL pH and Nugent score in women that got pregnant were −1.32 (−1.43–−1.21) and −3.90 (−4.25–−3.55), respectively, and in women that did not get pregnancy these reductions were only −0.19 (−0.29–−0.09) and−0.54 (−0.85–−0.23), respectively (Table 6; Appendix A).

The change in the vaginal cytokine concentrations after the probiotic treatment was similar to that described in the RA group: There was no modification in the vaginal TGF-β 1, TGF-β 2, and VEGF levels of women who did not become pregnant, but there was a mean (95% CI) significant increase of 2.29 (2.16–2.42) pg/mL, 1.25 (1.13–1.37) pg/mL, and 462 (411–513) pg/mL, respectively, in those who did (Table 6; Appendix A). In this INF group, there were already differences in the concentrations of TGF-β 2 and VEGF, but not in that of TGF-β 1, between those that became and those that did not became pregnant even before starting the treatment (Table 6).

The probiotic intervention resulted in a high degree of vaginal colonization by lactobacilli (6.46 [5.94–6.98] log_10_ CFU/mL) of all women that got pregnant, while this only happened in 46% of those that experienced a treatment failure, the density of lactobacilli reached being significantly lower (4.95 [4.28–5.62] log_10_ CFU/mL) (Table 6). Similarly to the RA group, the presence of *L. salivarius* (mean [95% CI] = 6.48 [6.28–6.68] copies/mL) was confirmed by qPCR in all women that got pregnant, but only in 31% of the women with unsuccessful pregnancies and, then, at a lower concentration (mean [95% CI] = 3.55 [3.24–3.86] copies/mL) (Table 6). The main difference in the lactobacilli profile of CVL samples of women in the INF group registered after the probiotic intervention was the detection of viable *L. salivarius* in all women who got pregnant, but only in 4 out of 13 of those women that failed to get pregnant. There were no differences in the metataxonomic profile at the genus level of CVL samples from women of the RA group regarding the pregnancy outcome (Figure 5; Appendix A).

### 3.7. Comparison of Vaginal Parameters between Women Who Became Pregnant and Those Who Did Not from Both the RA and INF Groups

The mean [95% CI] pH value in CVL samples was slightly but significantly more acidic in the women who become pregnant (5.69 [5.57–5.81] units) than in those who did not (5.99 [5.85–6.13] units) (*p* = 0.024; one-way ANOVA) (Figure 8; Appendix A). There were also differences in the concentration of vaginal cytokines TGF-β 2 and VEFG at the beginning of the study according to the final pregnancy outcome, but the differences were similar to those described already separately for RA and INF groups (Figure 8; Appendix A). The only parameters that did not differed initially between both groups were the Nugent score, TGF-β 1 concentration, and the frequency of detection and counts of lactobacilli (Figure 8; Appendix A). Globally, *Lactobacillus* was detected in all women who became pregnant, but only in half of those that did not (*p* < 0.001; Fisher exact probability test).

The probiotic intervention resulted in differential and remarkable changes in the vaginal parameters in those women who became pregnant but not in those who did not (Figure 8; Appendix A). First, the probiotic administration of *L. salivarius* CECT5713 resulted to be more effective regarding the change in the vaginal pH and Nugent score in women who got pregnant, which recorded mean (95% CI) decreases of −1.20 (−1.29–−1.12) and −3.56 (−3.82–−3.30) units, respectively (*p* = 0.000; one-way repeated measures ANOVA). In contrast, the change in these two parameters was smaller (−0.21 (−0.31–−0.10) and −0.58 (−0.88–−0.28) units, respectively) in the group of women who did not get pregnant (Figure 8; Appendix A). Second, the probiotic intervention led to a significant increase in the concentrations of vaginal cytokines TGF-β 1, TGF-β 2 and VEFG (mean [95% CI] increase of 1.76 [1.60–1.91] pg/mL, 1.25 [1.17–1.33] pg/mL, and 426 [378–473] pg/mL, respectively) in women who got pregnant but no change was registered in the group that did not (Figure 8; Appendix A). Third, regarding the lactobacilli profile of CVL samples, there was a mean (95% CI) increase of 2.67 (2.26–3.08) log_10_ CFU/mL units in viable *Lactobacillus* counts after the probiotic treatment in the group of women who became pregnant as opposed to those that did not. Differences were also noted on the *L. salivarius* content in CVL samples. This lactobacilli species was detected, and at a high concentration (mean [95% CI] = 6.70 [6.52–6.89] log_10_ copies/mL), in CVL samples from all women having a successful pregnancy unlike women who did not become pregnant (Figure 8; Appendix A). The metataxonomic profile at the genus level of CVL samples from women of the INF group was equal in women that did or did not become pregnant, except for a slightly higher relative frequency of *Escherichia/Shighella* in women that got pregnant (Figure 5; Appendix A).

### 3.8. Comparison of Vaginal Parameters between Control Women, All Women Who Became Pregnant and Those Who Did Not from Both RA and INF Groups

The analysis of post-intervention vaginal parameters (pH, Nugent score, TGF-β 1, TGF-β 2, VEGF, lactobacilli counts) revealed that the pH value of CVL samples and Nugent score in women who became pregnant after the probiotic intervention were similar to those of fertile control women (Table 7; Appendix A). The concentrations of TGF-β 1, TGF-β 2, and VEGF in post- intervention CVL samples of women who became pregnant were closer to those found in fertile control women, although statistically significant differences were found between them (Table 7; Appendix A). Besides, it is remarkable to note that the post-intervention concentration of VEGF in women that became pregnant was about twice that registered in fertile control women (mean [95% CI] = 755.0 [637.1–872.5] pg/mL and 406.0 [322.0–490.0] pg/mL, respectively). There was a high interindividual variation in lactobacilli counts varying from undetectable (in 57% of the women who did not become pregnant) to 7.5 log_10_ CFU/mL in CVL samples of women who did not become pregnant after the probiotic intervention, but the mean [95% CI] value (4.87 [3.83–5.90] log_10_ CFU/mL) was lower than in samples of the other participants (Table 7; Appendix A). There was less than 1 log_10_ CFU/mL difference between the lactobacilli viable counts in CVL samples of women who enjoyed a full term pregnancy after the probiotic intervention and those of fertile controls (mean [95% CI] = 6.47 [6.22–6.72] log_10_ CFU/mL and 7.24 [6.89–7.60] log_10_ CFU/mL, respectively) (Table 7; Appendix A).

Additionally, a network structure of the baseline vaginal bacterial genera communities on the three different groups of women (fertile controls, women who got pregnant after the probiotic intervention and women who did not get pregnant after the probiotic intervention) was constructed based on the genus-genus correlations (Figure 9). In the group of fertile women, the strongest correlation was observed between two minority genera, *Escherichia/Shigella* and *Enterococcus*; the most abundant genera, *Lactobacillus*, established negative and weak relationship with other Firmicutes (*Finegoldia* and *Peptoniphilus*) and *Prevotella.* In contrast, in the group of women with either repeated abortions or infertility of unknown origin, *Lactobacillus* showed strong negative association with two genera of the *Actinobacteria*, *Gardenella,* and *Bifidobacterium*. However, in the group of women that responded to the probiotic intervention and ended up in a successful pregnancy, the strongest negative association was between *Lactobacillus* and *Gardenella*, while in those women that did not get pregnant this negative association was weaker than that registered between *Lactobacillus* and *Bifidobacterium,* indicating that indeed the bacterial profile in CVL samples may indicate different fertility problems (Figure 9).

## 4. Discussion

In this study, the comparison between the vaginal microbiota of women with a history of reproductive failure, due to recurrent miscarriage or infertility, and healthy fertile women confirmed that dominance of specific species of *Lactobacillus* in the vaginal microbiota plays a determinant role in the success of human reproduction. Overall, the lowest vaginal pH values and Nugent scores were associated with vaginal communities dominated by lactobacilli, while those with the highest pH values and Nugent scores were associated with a depletion of lactobacilli. Close associations between low pH, low Nugent score and a high concentration and dominance of lactobacilli in the human vagina has been repeatedly reported [3,4,65]. In this study, the frequency of detection of lactobacilli in the vaginal samples was much higher in fertile women (100%) than in women with repetitive miscarriage (57%). Interestingly, infertile women showed the lowest percentage of women from whom lactobacilli could be isolated (26%). Use of antibiotics in both infancy and adulthood was significantly higher among women of the RA and INF groups than among women of the control group. It has been long known that opportunistic vaginal infections may arise as an adverse effect to the use of antibiotics because of their negative effect on the lactobacilli population [66]. The results obtained in this study suggest, for the first time, that an antibiotic-associated depletion of vaginal lactobacilli may have long-term health consequences by impairing fertility or embryo implantation and that such effect may be contrasted reversed by microbiological modulation of the vaginal ecosystem.

The species most frequently isolated from vaginal samples in this study belonged to *L. crispatus*, *L. gasseri*, *L. iners,* and *L. jensenii*, which are particularly common and abundant in the human vagina and absent or infrequently found in other human habitats [3,32,67]. Stable codominance of multiple *Lactobacillus* species is rarely observed in the same vaginal community [67]. Initial presence of *L. crispatus* seemed to be positively correlated with a successful reproductive outcome after the intervention with the probiotic assayed in this study. In contrast, initial presence of *L. iners* and *L. gasseri* seemed to be negatively correlated with a successful reproductive outcome after the probiotic intervention unless the *L. salivarius* strain provided in the trial was able to become dominant in the vaginal samples. *L. crispatus* and *L. iners* are probably the most common inhabitants of the healthy human vagina and are able to perform relevant ecological functions in the vaginal environment. Transitions from a vaginal community dominated by *L. iners* to one dominated by *L. crispatus*, and viceversa, seems to be relatively frequent [68]. The relationships between these two species and their potential functions have received an increasing scientific interest in the last years [67,68,69,70,71]. However, while there is a general agreement that a *L. crispatus*-dominated vaginotype promotes vaginal and reproductive health [72,73,74], the role of *L. iners* is very controversial since this peculiar species has been associated to beneficial roles for vaginal health [8,75] but, also, to dysbiosis, vaginal infections and a variety of gynecological conditions, including adverse pregnant outcomes [69,71,76,77,78]. Functional studies are required to investigate its roles in vaginal bacterial communities and whether, under certain circumstances, it can be used as a biomarker of reproductive failure.

A characterization of some properties of *L. salivarius* CECT5713 that may be relevant for vaginal and reproductive health showed that this strain was able to inhibit all the clinical isolates of *G. vaginalis*, *S. agalactiae*, *C. albicans*, *C. glabrata*, *C. parapsilosis*, and *U. urealyticum* tested in this study. This antimicrobial activity is relevant since vaginal infections are associated with an increased risk of adverse urogenital and reproductive health outcomes [79]. *L. salivarius* CECT5713 has a high acidifying ability by producing high amounts of L-lactic acid and small amounts of acetic acid [37]. Eubiosis and dysbiosis in the vaginal communities are distinguished by the high concentration of lactic acid and the high acidity that characterize the eubiosis state [79,80,81], as a direct result of the metabolic activity of the local lactobacilli, which is enough to inactivate reproductive tract pathogens, including viruses, bacteria and yeasts [49,82,83,84,85,86,87]. The capability and rate of production of lactic acid by lactobacilli is strain-specific and only high levels of lactic acid and a concomitant very low pH can inhibit microbial growth efficiently in the local vaginal biofilm [88,89]. From this point of view, *L. salivarius* CECT5713 seems a suitable candidate as a probiotic for the cervicovaginal target. In addition, this strain encodes an α-amylase in its genome (GenBank: ADJ79335.1), which is fully functional as revealed in the activity assays performed in this work. This enzyme might contribute, together with host α-amylase, to degradation of vaginal glycogen and, therefore, to increase lactic acid production and to maintain the vaginal pH at ≤4.5, promoting the desired lactobacilli dominance in the vaginal ecosystem [90].

Other properties of *L. salivarius* CECT5713 that are interesting in relation to the control of harmful vaginal microbes include a high rate of adhesion to vaginal cells and co-aggregation with the vaginal pathogens used in this study. High adherence of *L. salivarius* strains to vaginal cells has been previously observed and related to the prevention of vaginal colonization by *S. agalactiae* [49]. Both adhesion and co-aggregation activities seem to be highly strain-specific traits [48,49,91,92]. Cell-dependent reduction of *Candida* spp. adhesion by *Lactobacillus* species has been related to co-aggregation and competition for binding sites [93,94]. Overall, *L. salivarius* CECT5713 seems to be a strain suitable for applications involving vaginal homeostasis. This strain was isolated from human milk and infant feces of a healthy mother–child pair [37], and has been shown to be a good probiotic strain due to its extensive repertoire of desirable properties and safety, being particularly suited for application in the mother–infant dyad [38].

In this work, oral administration of *L. salivarius* CECT5713 to women of the RA and INF groups led to a relevant number of pregnancies. Women of such groups who had term pregnancies experienced significant changes in some key microbiological, biochemical and immunological parameters in the vaginal samples, such as concentration of cultivable lactobacilli, concentration of *L. salivarius* specific DNA, pH, Nugent score, and concentrations of VEGF, TGF-β 1 and TGF-β 2. The fact that all of them had high concentrations of *L. salivarius* in the vaginal samples and that DNA from this species was also detected by the qPCR assay reveals that the strain was able to reach and colonize the vaginal mucosa. The significant reductions of the pH values after the treatment indicate that the strain was metabolically active and suggests a good agreement between the in vitro potential of the strain and its in vivo capabilities.

The changes induced by *L. salivarius* CECT5713 in the concentrations of the growth factors VEGF, TGF-β 1 and TGF-β 2 seem to be particularly relevant and can be considered as biomarkers of the efficacy of the strain for the target pursued in the clinical trial. VEGF is a 45-kDa homodimeric heparin-binding glycoprotein with angiogenic activity that plays a key role as regulator of vasculogenesis, angiogenesis and vascular function in the human endometrium [95,96]. Vasculogenesis and angiogenesis are crucial steps for embryogenesis and particularly for embryo implantation (vessel formation and trophoblastic invasion) and both processes have been correlated with an increased expression of VEGF and VEGF receptors [97,98,99,100,101]; otherwise, endometrial angiogenesis may be impaired and result in a lethal phenotype, ranging from failed implantation to first-trimester miscarriage [95,102,103,104,105].

TGF-β 1 and TGF-β 2 also promote angiogenesis in vivo [106], and participate in implantation, trophoblast differentiation, and immunoregulation at the maternal-fetal interface [100,107]. Transcription of TGF-β 1 increases notably in human uterine endometrium during the first trimester of pregnancy [108], while recurrent pregnancy is associated with a decrease in the decidual TGF-β [109,110,111]. Expression of both VEGF and TGF-β 1 is highly regulated in a temporal and spatial manner during the early stages of implantation, a fact that underlines their critical role in the evolving pregnancy [109,110,111]. In addition, TGF-β 1 increases expression of VEGF in the trophoblast [111,112,113,114,115] suggesting a link between the action of both growth factors.

TGF-β 1 and TGF-β 2 are also of particular interest in this field because of their well-known roles in regulating the inflammatory response and inducing active immune tolerance in mucosal tissues [116,117]. Interestingly, both are present at very high concentrations in human seminal fluid [118,119], acting as male-female signaling agents that regulate the female immune response to sperm after coitus and promote maternal immune tolerance for embryo implantation and subsequent pregnancy [120,121,122,123]. Although studies in mouse models have shown that exposure to the high concentrations of TGF-β present in seminal fluid is absolutely required to boost uterine Treg cells prior to embryo implantation [124,125,126,127,128,129], this fact is not taken into account in many current ARTs, including IVF techniques, where such exposure is absent. Most TGF-β present in human semen is latent and requires activation to bind to receptors on cervical cells [130,131]. Interestingly, activation after coitus is facilitated by the acid pH of the vaginal environment [123] and, in this study, administration of *L. salivarius* CECT5713 led to an increase of TGF-β 1 and TGF-β 2 concentrations and, concomitantly, to a significant decrease in the vaginal pH values.

Our study has some limitations. First, the microbiota of the genitourinary tract of the partner was not evaluated and some studies have shown that male microbiota may also play a fundamental role in reproductive outcomes [132,133]. In fact, the couple (when applicable) should be considered as a single entity to achieve the best reproductive outcomes [134]. This approach will be taken into account in our future studies in this field. In addition, the metataxomomic analysis included in this study was carried at the genus level since the 16S rRNA gene approach has poor discriminatory power at the species level [135,136]. Other approaches, such as shotgun sequencing, should be used in the future to solve such limitation and to have a broader view of the vaginal microbiome.

Although our knowledge of the mechanisms that these early embryo–maternal interactions has increased in recent years, implantation remains as a rate-limiting step in human ART and the currently available treatments for infertility or recurrent pregnancy loss of unknown etiology have a rather limited efficacy [137,138]. Therefore, the possibility of enhancing angiogenic and tolerance activities in the endometrium by modifying the reproductive microbiota using bacterial strains specifically tailored for these targets provides a novel strategy to improve reproductive functions and deserves future basic and clinical research efforts.

## Figures and Tables

**Figure 1 nutrients-13-00162-f001:**
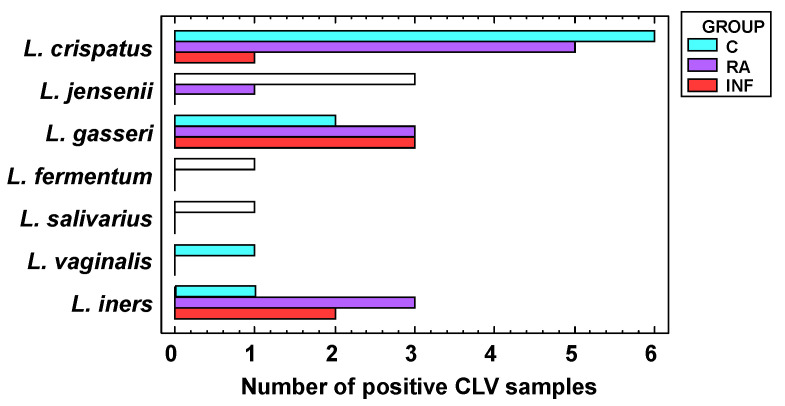
Dominant lactobacilli species (when lactobacilli could be isolated) in cervovaginal lavage (CVL) samples of fertile women (C, bluish green), women with repetitive abortion (RA, purple) and women with infertility of unknown origin (INF, red).

**Figure 2 nutrients-13-00162-f002:**
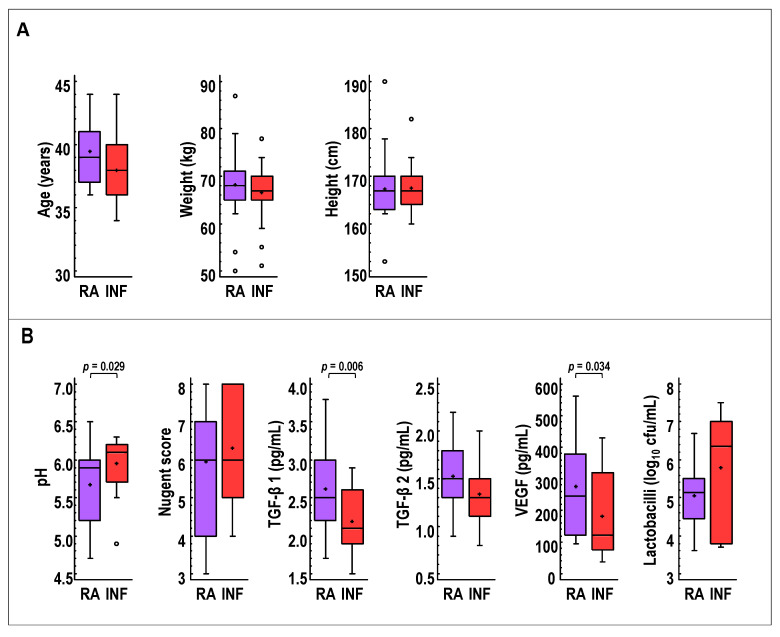
Comparison of selected baseline (**A**) demographic characteristics (age, weight and height) and (**B**) vaginal parameters (pH, Nugent score, TGF-β 1, TGF-β 2, and VEGF concentrations, and viable *Lactobacillus* counts) in CVL samples of women with repetitive abortion (RA, purple) and women with infertility of unknown origin (INF, red) at recruitment. For each boxplot, the line and the cross within the box represent the median and mean, respectively. The bottom and top boundaries of each box indicate the first and third quartiles (the 25th and 75th percentiles), respectively. The whiskers represent the lowest and highest values within the 1.5 interquartile range (IQR) and the dots outside the rectangles are suspected outliers (>1.5 × IQR). One-way ANOVA tests were used to compare both groups.

**Figure 3 nutrients-13-00162-f003:**
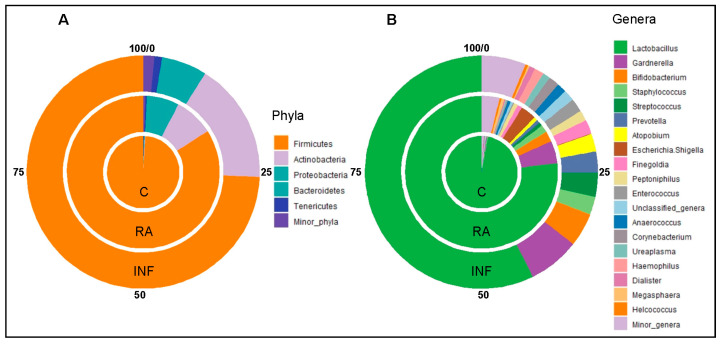
Pie charts showing the percentages of the relative abundances of the 5 most abundant phyla (**A**) and the 20 most abundant genera (**B**) in the CVL samples from healthy fertile women (inner pie charts; C group), women with a history of repetitive abortion (middle pie charts; RA group), and women with infertility of unknown origin (outer pie charts; INF group).

**Figure 4 nutrients-13-00162-f004:**
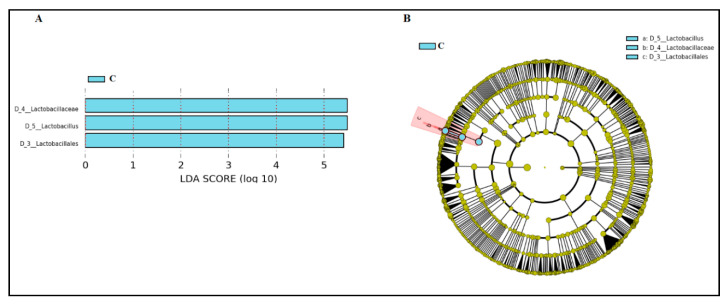
LEfSe analysis identifying taxonomic differences in the microbiota of CVL samples from healthy fertile women (C, bluish green) and women with repetitive abortion (RA) and with infertility of unknown origin (INF). Differentially abundant bacterial taxa were identified using linear discriminant analysis (LDA) and the effect size (LEfSe) algorithm. (**A**) Histogram of LDA scores (absolute LDA (log_10_) score > 2.0, *p* < 0.05) showing the substantial enrichment of *Lactobacillus* in the microbiota profile of the CVL samples from healthy fertile women. (**B**) Cladogram showing LEfSe comparison of differential bacterial taxa in CVL samples. The central point represents the root of the bacterial tree and each ring the next lower taxonomic level from phylum to genus (from the inner to the outer ring: phylum, class, order, family, and genus). The color node (other than yellow) indicates which taxa are significantly higher in relative abundance, and the diameter of the node is proportional to the relative abundance of the taxon.

**Figure 5 nutrients-13-00162-f005:**
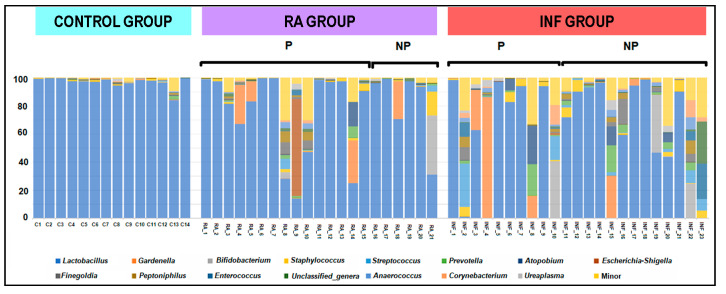
Relative abundance of the predominant bacterial genera in CVL samples of healthy fertile women (C), women with repetitive abortion (RA) and women with infertility of unknown origin (INF). In women with a history of reproductive failure, because either of recurrent miscarriage (RA group) or infertility (INF groups), P indicates the group of women who got pregnant after the probiotic intervention with *L. salivarius* CECT5713 and NP those women who did not.

**Figure 6 nutrients-13-00162-f006:**
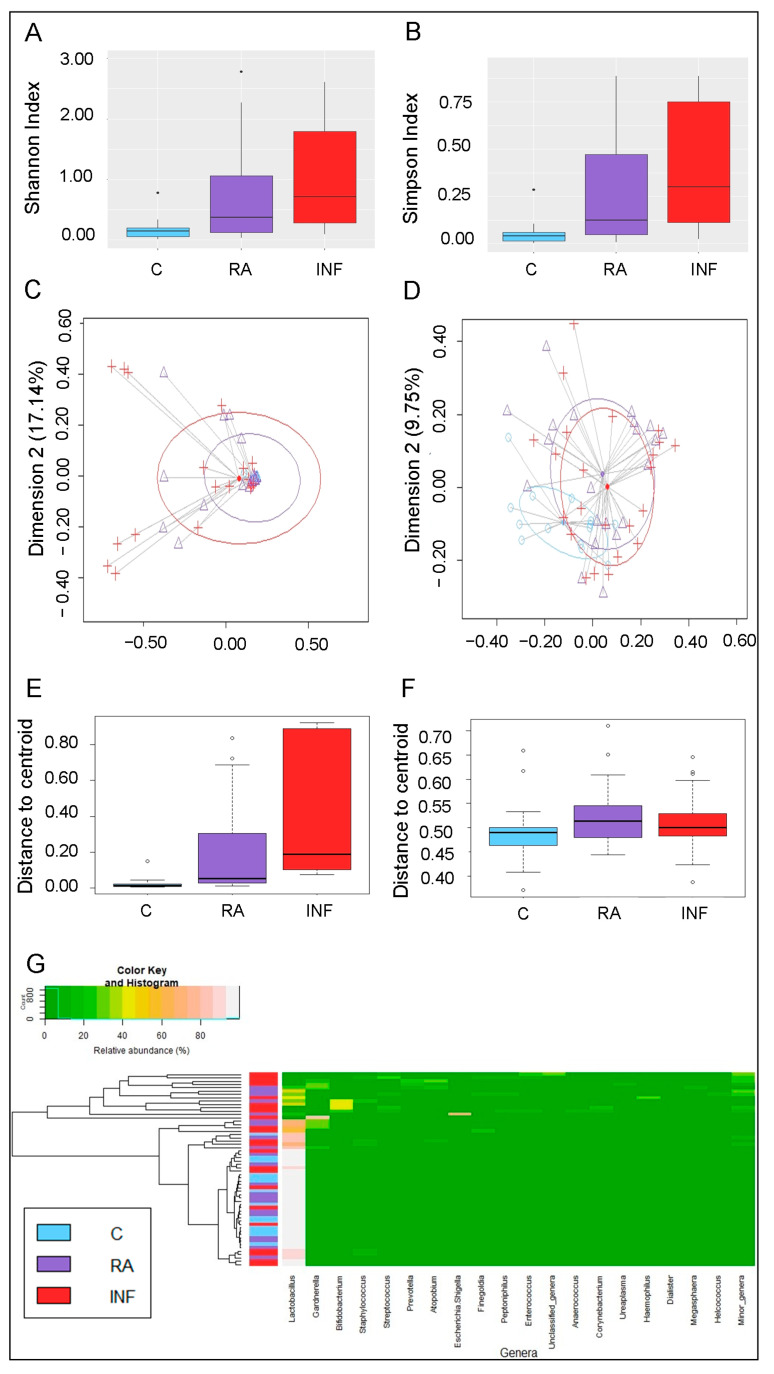
Metataxonomic profiles of CVL samples of healthy fertile women (C; bluish green), women with repetitive abortion (RA; purple) and women with infertility of unknown origin (INF; red). (**A**) Comparison of alpha diversity at genus level calculated using the Shannon index between the three groups of women. (**B**) Comparison of alpha diversity at genus level calculated using the Simpson index between the three groups of women. (**C**) Principal coordinate analysis (PCoA) plots of bacterial profiles at the genus level based on the Bray–Curtis dissimilarity analysis (relative abundance). (**D**) Principal coordinate analysis (PCoA) plots of bacterial profiles at the genus level based on the Jaccard’s coefficient for binary data (presence or absence). The values on each axis label in graphs C and D represent the percentage of the total variance explained by that axis. The differences between groups of CVL samples were analyzed using the PERMANOVA test with 999 permutations. (**E**) Comparison of the mean distances of samples to the centroids in the PCoA plots based on the Bray–Curtis dissimilarity index in each group. (**F**) Comparison of the mean distances of samples to the centroids in the PCoA plots based on the Jaccard’s coefficient (graph D) in each group. (**G**) Heatmap showing the relative abundance of the 20 most abundant bacterial genera (x axis) detected in CVL samples. The relative abundance of each bacterial genus within each sample is indicated by the color of the scale ranging from white (high relative abundance) to green (low relative abundance) as indicated in the scale shown at the left down corner. Dendrogram linkages are based upon relative abundance of the genus within the samples and *hclust* was used as the clustering algorithm. The column between the dendrogram of the vaginal samples and the individual values of the relative abundance of bacterial genera indicates the study group (control fertile women: C, in bluish green; women with repetitive abortion: RA, in purple; women with infertility of unknown origin: INF, in red). The differences between groups (C, healthy fertile women; RA, women with repetitive abortion; INF, women with infertility of unknown origin) were analyzed using Kruskal–Wallis tests with Bonferroni correction for data in panels A and B, and with one-way ANOVA tests for data in panels E and F.

**Figure 7 nutrients-13-00162-f007:**
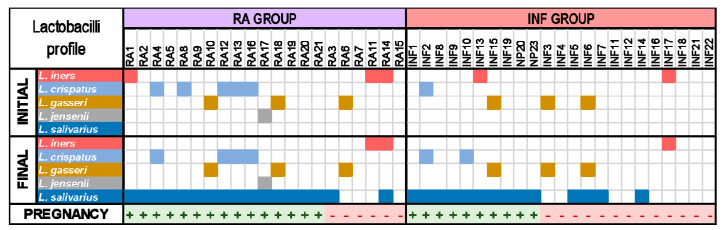
Changes in the profile of dominant *Lactobacillus* species in CVL samples from women with a history of repetitive abortion (RA group) and women with infertility of unknown origin (INF group) after the probiotic intervention with *L. salivarius* CECT5713. The outcome is indicated in the last file: +, successful full-term pregnancy and -, no pregnancy. The presence of isolates from a given species is indicated by a colored square.

**Figure 8 nutrients-13-00162-f008:**
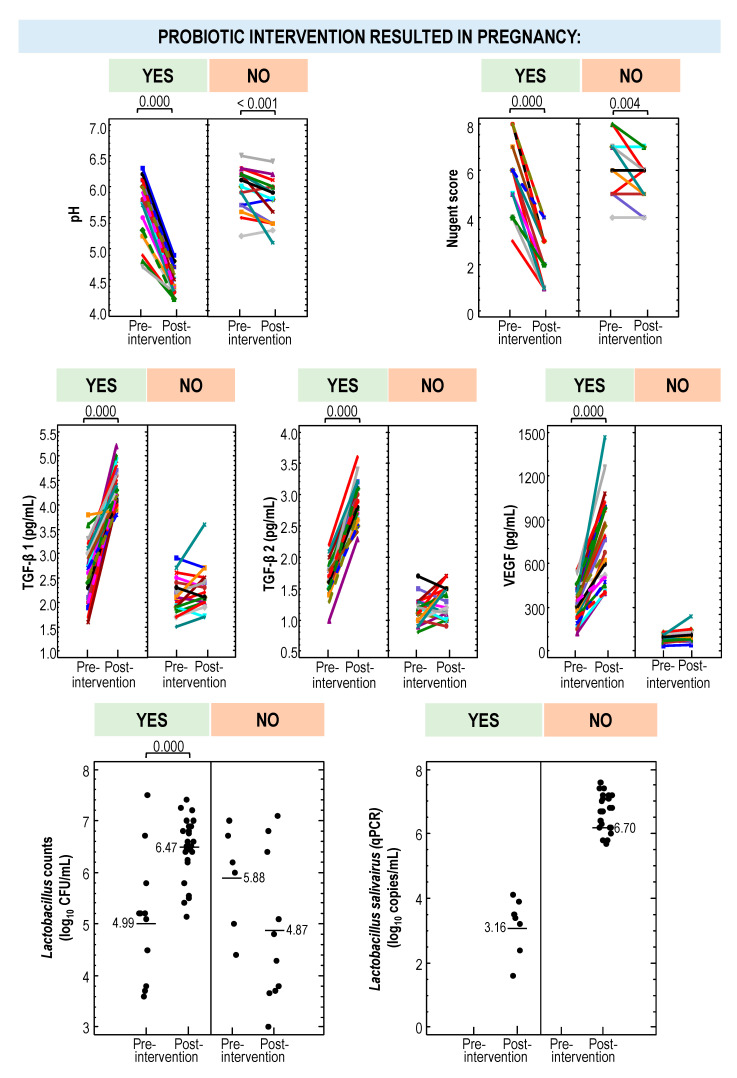
Changes in vaginal parameters (pH, Nugent score, TGF-β 1, TGF-β 2, and VEGF concentrations, viable *Lactobacillus* counts and *L. salivarius* copies in CVL samples) in women with a history of reproductive failure, because either of recurrent miscarriage (RA group) or infertility (INF groups), after the probiotic intervention with *L. salivarius* CECT5713 according to their outcome (pregnancy versus no pregnancy).

**Figure 9 nutrients-13-00162-f009:**
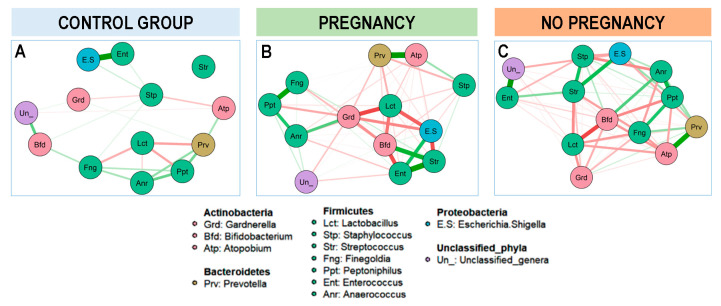
Estimated network structures based on a sample of 58 vaginal samples: 14 from healthy fertile women (**A**, Control group), 25 from women with a successful reproductive outcome after the probiotic intervention *L. salivarius* CECT5713 (**B**, Pregnancy) and 19 from women who did not have a successful pregnancy after the probiotic intervention with *L. salivarius* CECT5713 (**C**, No pregnancy). The 14 most abundant genera were represented. Red lines indicate negative correlation and green lines indicates positive correlation. The thickness and the intensity of the line reflects the intensity of the correlation.

**Table 1 nutrients-13-00162-t001:** Characteristics of the participants (*N* = 58) which included fertile women (Control group), women with a history of repetitive abortion (RA group), and women with infertility of unknown origin (INF group).

		Group		
Characteristic	Control (*n* = 14)	RA (*n* = 21)	INF (*n* = 23)	*p*-Value
**Age (years)**				
Mean (95% CI)	34.6 (33.5–35.8) ^a^	39.4 (38.5–40.4) ^b^	38.0 (37.1–38.9) ^b^	<0.001 ^2^
Range (min–max)	(28.0–45.0)	(36.0–44.0)	(34.0–44.0)	
**Weight (kg)**				
Mean (95% CI)	62.4 (59.7–65.0)	68.3 (66.1–70.4)	66.5 (64.5–68.6)	0.054 ^2^
Range (min–max)	(46.0–87.0)	(50.0–87.0)	(51.0–78.0)	
**Height (cm)**				
Mean (95% CI)	166 (164–168)	167 (165–169)	168 (166–169)	0.761 ^2^
Range (min–max)	(156–175)	(152–190)	(160–182)	
**Regularity of the menstrual cycle**				
Yes, *n* (%)	10 (71)	10 (48)	11 (48)	0.337 ^3^
No, *n* (%)	4 (29)	11 (52)	12 (52)	
**Duration of the menstrual cycle (days)**				
Mean (95% CI)	28.0 (27.4–28.7)	27.4 (26.9–27.9)	27.5 (27.0–28.0)	0.502 ^2^
Range (min–max)	(25.0–32.5)	(24.0–30.0)	(24.0–30.0)	
**History of infections**				
Vaginal, *n* (%)	2 (14)	13 (62)	8 (35)	0.017 ^3^
Urinary tract, *n* (%)	2 (14)	13 (62)	15 (65)	0.006 ^3^
Otorhinolaryngology, *n* (%)	3 (21)	13 (62)	12 (52)	0.057 ^3^
Lower respiratory tract, *n* (%)	2 (14)	7 (33)	7 (30)	0.490 ^3^
Skin, *n* (%)	1 (7)	3 (14)	4 (17)	0.800 ^3^
Gastrointestinal, *n* (%)	0 (0)	1 (5)	1 (4)	1.000
**Antibiotic usage ^1^**				
In infancy, *n* (%)	4 (29)	19 (90)	14 (61)	<0.001 ^3^
In adulthood, *n* (%)	4 (29)	16 (76)	19 (83)	0.003 ^3^
**History of other conditions**				
Allergies, *n* (%)	2 (14)	5 (24)	4 (17)	0.835 ^3^
Food intolerance, *n* (%)	0 (0)	8 (38)	13 (57)	0.001 ^3^
Thyroid disease, *n* (%)	0 (0)	5 (24)	3 (13)	0.125 ^3^

^1^ Antibiotic usage means ≥4 annual treatments due to recurrent infections. ^2^ One-way ANOVA tests were used to evaluate differences in mean values of women age, weight, and height and duration of the menstrual cycle between groups. Values followed by different superscript letters within the same row indicate statistically significant differences between groups according to Scheffé post hoc comparison tests. ^3^ Freeman–Halton extension of the Fisher exact probability tests for a 2 × 3 contingency table were used to compute the (two-tailed) probability of obtaining a distribution of values of categorical variables (regularity of the menstrual cycle, history of infections, antibiotic usage and history of other conditions).

**Table 2 nutrients-13-00162-t002:** Comparison of baseline vaginal parameters (pH, Nugent score, cytokines, and microbiology) of the participants (*n* = 58) which included fertile women (Control group), women with a history of repetitive abortion (RA group), and women with infertility of unknown origin (INF group).

		Group		
Vaginal Parameter	Control (*n* = 14)	RA (*n* = 21)	INF (*n* = 23)	*p*-Value
**pH**				
Mean (95% CI)	4.53 (4.38–4.68) ^a^	5.67 (5.55–5.79) ^b^	5.96 (5.84–6.07) ^b^	0.000 ^1^
Range (min–max)	(4.20–5.00)	(4.70–6.50)	(4.90–6.30)	
**Nugent score**				
Mean (95% CI)	1.79 (1.27–2.30) ^a^	5.95 (5.54–6.37) ^b^	6.30 (5.91–6.70) ^b^	0.000 ^1^
Range (min–max)	(0.00–4.00)	(3.00–8.00)	(4.00–8.00)	
**TGF-β 1**, pg/mL				
Mean (95% CI)	4.83 (4.65–5.01) ^a^	2.62 (2.47–2.76) ^b^	2.19 (2.05–2.33) ^c^	0.000 ^1^
Range (min–max)	(4.20–5.30)	(1.70–3.80)	(1.50–2.90)	
**TGF-β 2**, pg/mL				
Mean (95% CI)	3.22 (3.10–3.34) ^a^	1.52 (1.43–1.62) ^b^	1.33 (1.24–1.43) ^b^	0.000 ^1^
Range (min–max)	(2.70–3.70)	(0.90–2.20)	(0.80–2.00)	
**VEGF**, pg/mL				
Mean (95% CI)	406.0 (322.0–490.0) ^a^	274.8 (206.0–343.0) ^a,b^	181.2 (116.0–247.0) ^b^	0.016 ^1^
Range (min–max)	(1.4–929.0)	(95.0–562.0)	(38.0–431.0)	
**Lactobacilli**				
**Positive women**	14 (100)	12 (57)	6 (26)	<0.001 ^3^
**Viable counts**, log_10_ CFU/mL ^2^				
Mean (95% CI)	7.24 (6.89–7.60) ^a^	5.04 (4.66–5.42) ^b^	5.78 (5.24–6.32) ^b^	0.000 ^1^
Range (min–max)	(6.80–7.70)	(3.60–6.70)	(3.70–7.50)	
***L. salivarius* qPCR**, log_10_ copies/mL				
*n* (%)	1 (7)	0	0	
Mean (95% CI)	7.29			

^1^ One-way ANOVA tests were used to evaluate differences in mean values between groups. Values followed by different superscript letters within the same row indicate statistically significant differences between groups according to Scheffé post hoc comparison tests. ^2^ Mean (95% CI) and range (min–max) values in lactobacilli-positive women. ^3^ Freeman–Halton extension of the Fisher exact probability test for a 2 × 3 contingency table were used to compute the (two-tailed) probability of obtaining a distribution of values of lactobacilli positive women. Abbreviations: TGF-β 1, transforming growth factor β 1; TGF-β 2, transforming growth factor β 2; VEGF, vascular endothelial growth factor.

**Table 3 nutrients-13-00162-t003:** Relative frequencies, medians and interquartile range (IQR) of the most abundant bacterial phyla and genera detected in CVL samples from fertile women (Control group), women with a history of repetitive abortion (RA group), and women with infertility of unknown origin (INF group).

	Control (*n* = 14)		RA (*n* = 21)		INF (*n* = 23)		
Phylum Genus	*n* (%) ^1^	Median (IQR)	*n* (%)	Median (IQR)	*n* (%)	Median (IQR)	*p*-Value ^2^
***Firmicutes***	14 (100)	99.60 (99.18–99.80)	21 (100)	97.29 (72.34–99.35)	23 (100)	89.96 (52.46–98.85)	0.001
*Lactobacillus*	14 (100)	97.88 (96.92–99.31)	21 (100)	93.49 (67.18–97.53)	23 (100)	71.95 (0.76–94.09)	0.001
*Staphylococcus*	13 (93)	0.31 (0.11–0.66)	19 (90)	0.45 (0.03–1.51)	22 (96)	0.75 (0.14–5.40)	0.260
*Streptococcus*	9 (64)	0.02 (<0.01–0.03)	14 (67)	0.01 (<0.01–0.34)	16 (70)	0.06 (<0.01–2.04)	0.180
*Finegoldia*	13 (93)	0.17 (0.03–0.28)	18 (86)	0.16 (0.07–0.61)	17 (74)	0.12 (0.02–1.24)	0.760
*Peptoniphilus*	11 (79)	0.06 (0.01–0.21)	16 (76)	0.10 (0.02–0.49)	17 (74)	0.09 (<0.01–1.45)	0.670
*Enterococcus*	2 (14)	<0.01 (<0.01–<0.01)	6 (29)	<0.01 (<0.01–0.04)	12 (52)	0.01 (<0.01–0.19)	0.044
*Anaerococcus*	11 (79)	0.03 (0.01–0.16)	18 (86)	0.10 (0.05–0.30)	18 (78)	0.12 (0.01–1.71)	0.220
***Actinobacteria***	12 (86)	0.09 (0.02–0.20)	21 (100)	0.32 (0.08–7.87)	23 (100)	4.84 (0.1–34.36)	0.012
*Gardnerella*	4 (29)	<0.01 (<0.01–0.01)	11 (52)	0.01 (<0.01–0.12)	9 (39)	<0.01 (<0.01–0.04)	0.300
*Bifidobacterium*	3 (21)	<0.01 (<0.01–<0.01)	9 (43)	<0.01 (<0.01–0.07)	9 (39)	<0.01 (<0.01–0.03)	0.300
*Atopobium*	2 (14)	<0.01 (<0.01–<0.01)	7 (33)	<0.01 (<0.01–0.01)	13 (57)	0.02 (<0.01–0.12)	0.015
***Proteobacteria***	1 (93)	0.07 (0.02–0.10)	21 (100)	0.28 (0.09–0.69)	22 (96)	0.23 (0.09–0.64)	0.003
*Escherichia/Shigella*	1 (7)	<0.01 (<0.01–<0.01)	9 (43)	<0.01 (<0.01–0.02)	8 (35)	<0.01 (<0.01–0.01)	0.084
***Bacteroidetes***	10 (71)	0.03 (<0.01–0.08)	18 (86)	0.16 (0.06–1.33)	22 (96)	0.80 (0.05–3.19)	0.006
*Prevotella*	8 (57)	0.02 (<0.01–0.08)	15 (71)	0.06 (<0.01–0.45)	19 (83)	0.70 (0.01–2.55)	0.660
*Tenericutes*	6 (43)	<0.01 (<0.01–0.16)	5 (24)	<0.01 (<0.01–<0.01)	10 (43)	<0.01 (<0.01–0.97)	0.290
**Minor phyla**	14 (100)	0.13 (0.07–0.18)	21 (100)	0.16 (0.07–0.65)	23 (100)	0.17 (0.09–1.29)	0.280
Minor genera	14 (100)	0.30 (0.09–0.70)	21 (100)	0.91 (0.27–2.54)	23 (100)	2.26 (0.40–8.35)	0.038
Unclassified_genera	14 (100)	0.09 (0.05–0.12)	21 (100)	0.13 (0.07–0.66)	23 (100)	0.14 (0.04–0.36)	0.170

^1^*n* (%): Number of samples in which the phylum/genus was detected (relative frequency of detection). ^2^ Kruskal–Wallis rank tests with Bonferroni correction.

**Table 4 nutrients-13-00162-t004:** Main outcomes after the probiotic treatment with *L. salivarius* CECT5713 in women with repetitive abortion (RA) and women with infertility of unknown origin (INF).

	Group		Total	Ratio (95% CI)
Outcome	RA	INF	RA + INF	(RA/INF)
Pregnancy (events/total events)	17/21	12/23	29/44	
Pregnancy effectiveness (95% CI)	81% (64–98%)	52% (32–73%)	66% (52–80%)	1.55 (1.00–2.42)
Successful pregnancy ^1^ (events/total events)	15/21	10/23	25/44	
Reproductive success (95% CI)	71% (52–91%)	43% (23–64%)	57% (42–72%)	1.64 (0.96–2.82)

^1^ Two women in each group end up in abortion.

**Table 5 nutrients-13-00162-t005:** Effect of the probiotic intervention with *L. salivarius* CECT5713 on the vaginal parameters of women who were able to complete a full-term pregnancy (*n* = 15) and of those who did not (*n* = 6) among the women that had a history of repetitive abortion (RA group; *n* = 21).

	Probiotic Intervention Resulted in Pregnancy		
	Yes (*n* = 15)	No (*n* = 6)	
Vaginal Parameter	Mean (95% CI)	Mean (95% CI)	*p*-Value ^1^
**pH**			
Baseline	5.58 (5.39–5.77)	5.88 (5.58–6.18)	0.221
Post-intervention	4.45 (4.34–4.57)	5.65 (0.13–5.46)	0.000
Change	−1.13 (−1.27–−0.99)	−0.23 (−0.45–−0.01)	<0.001
*p*-value ^3^	0.000	0.002	
**Nugent score**			
Baseline	5.87 (5.24–6.49)	6.17 (5.18–7.15)	0.708
Post-intervention	2.53 (2.13–2.94)	5.50 (4.86–6.14)	0.000
Change	−3.33 (−3.73–−2.93)	−0.67 (−1.29–−0.04)	0.000
*p*-value ^3^	0.000	0.102	
**TGF-β 1**, pg/mL			
Baseline	2.81 (2.62–3.00)	2.15 (1.85–2.45)	0.014
Post-intervention	4.21 (4.05–4.36)	2.47 (2.22–2.71)	0.000
Change	1.40 (1.18–1.62)	0.32 (−0.02–0.66)	<0.001
*p*-value ^3^	0.000	0.098	
**TGF-β 2**, pg/mL			
Baseline	1.67 (1.57–1.78)	1.15 (0.99–1.31)	<0.001
Post-intervention	2.93 (2.81–3.05)	1.30 (1.11–1.49)	0.000
Change	1.25 (1.12–1.38)	0.15 (−0.05–0.35)	0.000
*p*-value ^3^	0.000	0.328	
**VEGF**, pg/mL			
Baseline	341 (300–382)	109 (44–173)	<0.001
Post-intervention	743 (640–846)	138 (−25–301)	<0.001
Change	402 (319–485)	29 (−102–160)	0.002
*p*-value ^3^	0.000	0.189	
**Lactobacilli presence**, *n* (%)			
Baseline	9 (60)	3 (50)	0.523 ^2^
Post-intervention	15 (100)	4 (67)	0.071 ^2^
Change	6 (40)	1 (17)	0.613 ^2^
**Lactobacilli counts**, log_10_ CFU/mL			
Initial	4.99 (4.48–5.50)	5.20 (4.31–6.09)	0.752
Final	6.52 (6.22–6.81)	4.74 (4.17–5.31)	<0.001
Change	2.44 (1.84–3.04)	0.16 (−0.99–1.32)	0.019
*p*-value ^3^	<0.001	0.697	
***L. salivarius* qPCR**, *n* (%)			
Initial	nd	nd	
Final	15 (100)	3 (50)	0.015 ^2^
***L. salivarius* qPCR**, log_10_ copies/mL ^4^			
Initial	-	-	
Final	6.85 (6.58–7.12)	2.63 (0.41–3.24)	<0.000

^1^ One-way ANOVA tests were used to evaluate differences in mean values between groups, except for lactobacilli presence. ^2^ Fisher exact probability test for a 2 × 2 contingency table. ^3^ One-way repeated measures ANOVA tests were used to determine whether there was a change in each group of participants when comparing the baseline and post-intervention parameters. ^4^ Mean (95% CI) of *L. salivarius* qPCR (copies/mL) in positive samples.

**Table 6 nutrients-13-00162-t006:** Effect of the probiotic intervention with *L. salivarius* CECT5713 on the vaginal parameters of women who were able to complete a full-term pregnancy (*n* = 15) and of those who did not (*n* = 6) among the women with infertility of unknown origin (INF group; *n* = 23).

	Probiotic Intervention Resulted in Pregnancy		
	Yes (*n* = 15)	No (*n* = 6)	
Vaginal Parameter	Mean (95% CI)	Mean (95% CI)	*p*-Value ^1^
**pH**			
Baseline	5.85 (5.70–6.00)	6.04 (5.58–6.17)	0.190
Post-intervention	4.53 (4.42–4.64)	5.85 (5.75–5.95)	0.000
Change	−1.32 (−1.43–−1.21)	−0.19 (−0.29–−0.09)	0.000
*p*-value ^3^	0.000	0.002	
**Nugent score**			
Baseline	6.00 (5.40–6.60)	6.54 (6.01–7.07)	0.334
Post-intervention	2.10 (1.61–2.59)	6.00 (5.57–6.43)	0.000
Change	−3.90 (−4.25–−3.55)	−0.54 (−0.85–−0.23)	0.000
*p*-value ^3^	0.000	0.028	
**TGF-β 1**, pg/mL			
Baseline	2.29 (2.10–2.48)	2.11 (1.94–2.28)	0.308
Post-intervention	4.58 (4.41–4.75)	2.18 (2.04–2.33)	0.000
Change	2.29 (2.16–2.42)	0.08 (−0.04–0.19)	0.000
*p*-value ^3^	0.000	0.281	
**TGF-β 2**, pg/mL			
Baseline	1.56 (1.46–1.66)	1.16 (1.07–1.25)	<0.001
Post-intervention	2.81 (2.68–2.94)	1.26 (1.15–1.38)	0.000
Change	1.25 (1.13–1.37)	0.10 (<−0.01–0.20)	0.000
*p*-value ^3^	0.000	0.203	
**VEGF**, pg/mL			
Baseline	311 (279–343)	81 (53–109)	0.000
Post-intervention	773 (695–850)	87 (19–155)	0.000
Change	462 (411–513)	6 (−39–50)	0.000
*p*-value ^3^	0.000	0.165	
**Lactobacilli presence**, *n* (%)			
Baseline	3 (30)	3 (23)	0.537 ^2^
Post-intervention	10 (100)	6 (46)	0.007 ^2^
Change	7 (70)	3 (23)	0.040 ^2^
**Lactobacilli counts**, log_10_ CFU/mL			
Initial	5.00 (3.22–6.78)	6.57 (4.78–8.35)	0.290
Final	6.46 (5.94–6.98)	4.95 (4.28–5.62)	0.017
Change	3.05 (2.45–3.64)	0.32 (−0.46–1.09)	<0.001
*p*-value ^3^	<0.001	0.451	
***L. salivarius* qPCR**, *n* (%)			
Initial	nd	nd	
Final	10 (100)	4 (31)	0.002 ^2^
***L. salivarius* qPCR**, log_10_ copies/mL ^4^			
Initial	-	-	
Final	6.48 (6.28–6.68)	3.55 (3.24–3.86)	0.000

^1^ One-way ANOVA tests were used to evaluate differences in mean values between groups, except for lactobacilli presence. ^2^ Fisher exact probability test for a 2 × 2 contingency table. ^3^ One-way repeated measures ANOVA tests were used to determine whether there was a change in each group of participants when comparing the baseline and post-intervention parameters. ^4^ Mean (95% CI) of *L. salivarius* qPCR (copies/mL) in positive samples.

**Table 7 nutrients-13-00162-t007:** Comparison of vaginal parameters (pH, Nugent score, TGF-β 1, TGF-β 2, and VEGF concentrations, and *Lactobacillus* counts) of all women who were able to complete a full-term pregnancy (*n* = 25) and of those who did not (*n* = 19) among all women with a history of repetitive abortion and with infertility of unknown origin (RA and INF groups) after the probiotic intervention with *L. salivarius* CECT5713 and vaginal parameters of fertile women (Control group; *n* = 14).

		Probiotic Intervention Resulted in Pregnancy		
Vaginal Parameter	Control (*n* = 14) Mean (95% CI)	Yes (*n* = 25) Mean (95% CI)	No (*n* = 23) Mean (95% CI)	*p*-Value
pH	4.53 (4.38–4.68) ^a^	4.48 (4.39–4.58) ^a^	5.78 (5.62–5.95) ^b^	0.000 ^1^
Nugent score	1.79 (1.27–2.30) ^a^	2.36 (1.92–2.80) ^a^	5.84 (5.35–6.33) ^b^	0.000 ^1^
TGF-β 1, pg/mL	4.83 (4.65–5.01) ^a^	4.36 (4.20–4.52) ^b^	2.27 (2.06–2.48) ^c^	0.000 ^1^
TGF-β 2, pg/mL	3.22 (3.10–3.34) ^a^	2.88 (2.75–3.01) ^b^	1.27 (1.15–1.40) ^c^	0.000 ^1^
VEGF, pg/mL	406.0 (322.0–490.0) ^a^	755.0 (637.1–872.5) ^b^	103.3 (82.4–124.1) ^c^	0.000 ^1^
Lactobacilli				
Positive women, *n* (%)	14 (100)	25 (100)	10 (43)	<0.001 ^2^
Viable counts ^3^, log_10_ CFU/mL	7.24 (6.89–7.60) ^a^	6.47 (6.22–6.72) ^b^	4.87 (3.83–5.90) ^c^	0.000 ^1^

^1^ One-way ANOVA tests were used to evaluate differences in mean values between groups. Values followed by different superscript letters within the same row indicate statistically significant differences between groups according to Scheffé post hoc comparison tests. ^2^ Freeman–Halton extension of the Fisher exact probability tests for a 2 × 3 contingency table were used to compute the (two-tailed) probability of obtaining a distribution of values of lactobacilli positive women. ^3^ Mean (95% CI) of *L. salivarius* qPCR (copies/mL) in lactobacilli-positive women. TGF-β 1, transforming growth factor-β 1; TGF-β 2, transforming growth factor-β 2; VEGF, vascular endothelial growth factor.

## Data Availability

The data presented in this study are available on request from the corresponding author. The data are not publicly available due to ethical restrictions.

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
