# Peer review of "Application of Ligilactobacillus salivarius CECT5713 to Achieve Term Pregnancies in Women with Repetitive Abortion or Infertility of Unknown Origin by Microbiological and Immunological Modulation of the Vaginal Ecosystem"

_nutrients, 2021, doi:10.3390/nu13010162_

Round 1

Reviewer 1 Report

This study, by comparing the vaginal microbiota of women with a history of reproductive failure and healthy fertile women, confirmed that the predominance of specific Lactobacillus species in the vaginal microbiota plays a decisive role in the success of human reproduction.

However, it would be good to mention in the discussion the importance of an evaluation of the male microbiota which also plays a fundamental role in reproduction:the simultaneous evaluation of the microbiota of both partners (Tomaiuolo R, Veneruso I, Cariati F, D'Argenio V. Microbiota and Human Reproduction: The Case of Female Infertility. High Throughput. 2020 May 3;9(2):12; Tomaiuolo R, Veneruso I, Cariati F, D'Argenio V. Microbiota and Human Reproduction: The Case of Male Infertility. High Throughput. 2020 Apr 13;9(2):10. doi: 10.3390/ht9020010) should be taken into account considering their mutual influences and the potential effects on couples' and offspring health. The couple should be considered as a single entity to achieve the best reproductive outcome (Cariati F, D'Argenio V, Tomaiuolo R. The evolving role of genetic tests in reproductive medicine. J Transl Med. 2019 Aug 14;17(1):267. doi: 10.1186/s12967-019-2019-8).

Author Response

Question 1. However, it would be good to mention in the discussion the importance of an evaluation of the male microbiota which also plays a fundamental role in reproduction:the simultaneous evaluation of the microbiota of both partners (Tomaiuolo R, Veneruso I, Cariati F, D'Argenio V. Microbiota and Human Reproduction: The Case of Female Infertility. High Throughput. 2020 May 3;9(2):12; Tomaiuolo R, Veneruso I, Cariati F, D'Argenio V. Microbiota and Human Reproduction: The Case of Male Infertility. High Throughput. 2020 Apr 13;9(2):10. doi: 10.3390/ht9020010) should be taken into account considering their mutual influences and the potential effects on couples' and offspring health. The couple should be considered as a single entity to achieve the best reproductive outcome (Cariati F, D'Argenio V, Tomaiuolo R. The evolving role of genetic tests in reproductive medicine. J Transl Med. 2019 Aug 14;17(1):267. doi: 10.1186/s12967-019-2019-8).

Answer: Thank you for your comment. We agree that evaluation of the male microbiota is also very relevant for reproductive outcomes. Therefore, we have add the following sentence in the discussion section (p. 24, lines 791-795). “Our study has some limitations. First, the microbiota of the genitourinary tract of the partner was not evaluated and some studies have shown that male microbiota may also play a fundamental role in reproductive outcomes [132,133]. In fact, the couple (when applicable) should be considered as a single entity to achieve the best reproductive outcomes [134]. This approach will be taken into account in our future studies in this field.

The references suggested by the reviewer have been included in this paragraph since the three of them are very relevant to the field.

Reviewer 2 Report

In their manuscript entitled” Application of Ligilactobacillus salivarius CECT5713 to achieve term pregnancies in women with repetitive abortion or infertility of unknown origin by microbiological and immunological modulation of the vaginal ecosystem” Fernandez et al. investigated the administration of the Ligilactobacillus salivarius CECT5713 in order to evaluate its ability in increasing the pregnancy rate in RA and INF women.

The topic is interesting and the impact on clinical knowledge could be relevant.

Major points:

  • The “Specie level” should have been included (in Figure 3)
  • The authors should show the LSFe plot showing metagenomics species that are overexpressed and under-expressed in the 3 cohorts examined.

Minor points:

  • The introduction. The AA could include more recent literature. Here some examples:
  • Riganelli et al. (2020) Structural Variations of Vaginal and Endometrial Microbiota: Hints on Female Infertility
  • Haahr, T. et al. (2016) Abnormal vaginal microbiota may be associated with poor reproductive outcomes: a prospective study in IVF patients
  • Peric, A., Weiss, J., Vulliemoz, N., Baud, D., and Stojanov, M. (2019). Bacterial colonization of the female upper genital tract.
  • The Table 3 is not clear. Could the AA replace the table with a classic pie chart for each category (Phylum; Family; Species)
  • The Figure 4 needs a re-make up: the panels A/B/E/F are too big and the panels C/D are too small and not readable
  • Could the AA indicate the general rate of the infertility of the 2 cohorts (RA /INF) as reported by their Institute?

Author Response

Question 1. The “Specie level” should have been included (in Figure 3)

Answer: The 16S rRNA gene has been a mainstay of sequence-based bacterial analysis for decades. However, the final metatoxonomic assignment in the high throughput sequencing can be affected by the choice of the 16S rRNA region on the bacterial community metabarcoding, specially at the species level identification [1,2]. Therefore, we chose the general-level  as the lowest taxonomical level in this study in order to , in our study, in order to avoid bias at the species-level metataxonomic identification associated with the DNA fragment length (2×300 nt) and the 16S rRNA hypervariable region (V3-V4 region).

In addition, our classifier pipeline uses a Naïve Bayesian algorithm to sort the sequences and, to avoid significant mistakes, it is recommended to use a taxonomic level higher than the species level [3]. As an example, the metataxomomic approach is suitable to detect sequences belonged to the genus Lactobacillus but it is unable to distinguish between those 16S rDNA fragments corresponding to L. crispatus or L. iners.

  1. Graspeuntner, S., Loeper, N., Künzel, S., Baines, J. F. & Rupp, J. Selection of validated hypervariable regions is crucial in 16S-based microbiota studies of the female genital tract. Scientific Reports 8, 9678 (2018)
  2. Bukin, Y., Galachyants, Y., Morozov, I. et al. The effect of 16S rRNA region choice on bacterial community metabarcoding results. Sci Data 6, 190007 (2019). https://doi.org/10.1038/sdata.2019.7
  3. Yarza, P., Yilmaz, P., Pruesse, E. et al. Uniting the classification of cultured and uncultured bacteria and archaea using 16S rRNA gene sequences. Nat Rev Microbiol 12, 635–645 (2014). https://doi.org/10.1038/nrmicro3330

There are many other references in the literature describing this problem that seems inherent to the 16S rRNA approach:

Milani C, Alessandri G, Mangifesta M, Mancabelli L, Lugli GA, Fontana F, Longhi G, Anzalone R, Viappiani A, Duranti S, Turroni F, Costi R, Annicchiarico A, Morini A, Sarli L, Ossiprandi MC, van Sinderen D, Ventura M. Untangling Species-Level Composition of Complex Bacterial Communities through a Novel Metagenomic Approach. mSystems. 2020 Jul 28;5(4):e00404-20. doi: 10.1128/mSystems.00404-20. 

Srinivasan, R.; Karaoz, U.; Volegova, M.; MacKichan, J.; Kato-Maeda, M.; Miller, S.; Nadarajan, R.; Brodie, E.L.; Lynch, S.V. Use of 16S rRNA gene for identification of a broad range of clinically relevant bacterial pathogens. PLoS ONE 2015, 10, e0117617. [CrossRef]

Mignard, S.; Flandrois, J.P. 16S rRNA sequencing in routine bacterial identification: A 30-month experiment. J. Microbiol. Methods 2006, 67, 574–581.

Větrovský T, Baldrian P. The variability of the 16S rRNA gene in bacterial genomes and its consequences for bacterial community analyses. PLoS One. 2013;8(2):e57923. doi: 10.1371/journal.pone.0057923. 

Winand, R., Bogaerts, B., Hoffman, S., Lefevre, L., Delvoye, M., Braekel, J. V., Fu, Q., Roosens, N. H., Keersmaecker, S. C., & Vanneste, K. Targeting the 16S rRNA gene for bacterial identification in complex mixed samples: comparative evaluation of second (Illumina) and third (Oxford nanopore technologies) generation sequencing technologies. Int J Mol Sci. 2019 Dec 31;21(1):298. doi: 10.3390/ijms21010298. 

In order to cope with the reviewer’s comment, we have add the following paragraph in the revised manuscript (p. 24, lines 791-798): “Our study has some limitations. (…). In addition, the metataxomomic analysis included in this study was carried at the genus level since the 16S rRNA gene approach has poor discriminatory power at the species level [135,136]. Other approaches, such as shotgun sequencing, should be used in the future to solve such limitation and to have a broader view of the vaginal microbiome.

Two new references to support this sentence:

135. Mignard, S.; Flandrois, J.P. 16S rRNA sequencing in routine bacterial identification: A 30-month experiment. J. Microbiol. Methods 2006, 67, 574–581. doi: 10.1016/j.mimet.2006.05.009.

136. Bukin, Y.S.; Galachyants, Y.P.; Morozov, I.V.; Bukin, S.V.; Zakharenko, A.S.; Zemskaya, T.I. The effect of 16S rRNA region choice on bacterial community metabarcoding results. Sci. Data 2019, 6, 190007. doi: 10.1038/sdata.2019.7.

Question 2. The authors should show the LSFe plot showing metagenomics species that are overexpressed and under-expressed in the 3 cohorts examined.

Answer: As suggested by the reviewer, we have performed the LEfSe analysis. A new figure showing the LDA scores and the cladogram of the differentially abundant bacterial taxa has been included in the revised manuscript (Figure 4). The information about how the LEfSe analysis was performed has been included in Section 2.8. Bioinformatic analysis (page 5, lines 228-231):

“The relative abundance values of the different bacterial taxa in the three groups of CVL samples (control, RA and INF) were analyzed using the linear discriminant analysis (LDA) effect size (LEfSe) algorithm [63] in an online version (http://huttenhower.sph.harvard.edu/galaxy/).”

New reference 63:

Segata, N., Izard, J., Waldron, L.et al. Metagenomic biomarker discovery and explanation. Genome Biol 12, R60 (2011). https://doi.org/10.1186/gb-2011-12-6-r60

Question 3. The introduction. The AA could include more recent literature. Here some examples:

Riganelli et al. (2020) Structural Variations of Vaginal and Endometrial Microbiota: Hints on Female Infertility

Haahr, T. et al. (2016) Abnormal vaginal microbiota may be associated with poor reproductive outcomes: a prospective study in IVF patients

Peric, A., Weiss, J., Vulliemoz, N., Baud, D., and Stojanov, M. (2019). Bacterial colonization of the female upper genital tract.

Answer: We agree that the suggested references are relevant to the field. In fact, the reference  Haahr, T. et al. (2016) was already included in our original manuscript. The other two references have been included in the introduction section of the revised manuscript (references 29 and 30).

Question 4. The Table 3 is not clear. Could the AA replace the table with a classic pie chart for each category (Phylum; Family; Species)

Answer: In the revised version of the manuscript we have included a classic pie chart (Figure 3) to represent the differences in the percentages of the relative abundances of the bacterial assignations of the reads at the phylum and genus level. This type of graph is constructed with the mean values of the relative abundance of the bacterial taxa (phylum, genus). As the distribution of these values do not follow a normal distribution we have considered convenient to maintain Table 3 in the manuscript because, although it requires some time to decipher, it gives more valuable information about the bacterial profile of the CVL samples of the three groups.

Question 5. The Figure 4 needs a re-make up: the panels A/B/E/F are too big and the panels C/D are too small and not readable

Answer: Figure 4 (Figure 6 in the revised manuscript) has been modified according to the reviewer’s suggestion. Panels C and D had been enlarged to be more easily readable.

Question 6. Could the AA indicate the general rate of the infertility of the 2 cohorts (RA /INF) as reported by their Institute?

Answer: The following information has been included in the revised manuscript (page 3, lines 100-101):

“(…) to increase pregnancy rates (currently ~29% after IVF procedures in this setting)”. This percentage includes all women with infertility of unknown origin (this is the only value available).

Round 2

Reviewer 2 Report

The AA accomplished all the requests